# MicroRNA-146a acts as a guardian of the quality and longevity of hematopoietic stem cells in mice

**Jimmy L Zhao[1]\*, Dinesh S Rao[2], Ryan M O'Connell[3], Yvette Garcia-Flores[1], David Baltimore[1]\***

[1]Department of Biology, California Institute of Technology, Pasadena, United States; [2]Department of Pathology and Laboratory Medicine, University of California, Los Angeles, Los Angeles, United States; [3]Department of Pathology, University of Utah, Salt Lake City, United States

**Abstract** During inflammation and infection, hematopoietic stem and progenitor cells are stimulated to proliferate and differentiate into mature immune cells, especially of the myeloid lineage. MicroRNA-146a (miR-146a) is a critical negative regulator of inflammation. Deletion of miR-146a produces effects that appear as dysregulated inflammatory hematopoiesis, leading to a decline in the number and quality of hematopoietic stem cells (HSCs), excessive myeloproliferation, and, ultimately, to HSC exhaustion and hematopoietic neoplasms. At the cellular level, the defects are attributable to both an intrinsic problem in the miR-146a–deficient HSCs and extrinsic effects of lymphocytes and nonhematopoietic cells. At the molecular level, this involves a molecular axis consisting of miR-146a, signaling protein TRAF6, transcriptional factor NF-κB, and cytokine IL-6. This study has identified miR-146a to be a critical regulator of HSC homeostasis during chronic inflammation in mice and provided a molecular connection between chronic inflammation and the development of bone marrow failure and myeloproliferative neoplasms.

**\*For correspondence:**
jzhao@caltech.edu (JLZ);
baltimo@caltech.edu (DB)

## Introduction

Hematopoietic stem cells (HSCs) have the ability to self-renew and replenish the entire hematopoietic repertoire during the lifetime of an organism. Balanced self-renewal vs differentiation of HSCs is intricately regulated to ensure the long-term maintenance of HSCs and the hematopoietic system (*Seita and Weissman, 2010*). Under stressed conditions, such as inflammation and infection, the balance is shifted in favor of hematopoietic stem and progenitor cell (HSPC) proliferation and differentiation to produce more mature immune cells (*King and Goodell, 2011*). Following the discovery that HSCs express TLRs and may sense and respond to infection and inflammatory signals directly (*Nagai et al., 2006*), there has been an increasing appreciation of the role of proinflammatory cytokines and infection in the modulation of HSC activity. Numerous recent studies have shown that TLR activation or interferon stimulation leads to proliferation, skewed myeloid differentiation and impaired engraftment, and self-renewal of HSCs (*Essers et al., 2009*; *Baldridge et al., 2010*; *Esplin et al., 2011*).

Since its discovery over 25 years ago, NF-κB has been shown to be active in a wide variety of innate and adaptive immune cells as well as nonhematopoietic cells and to function as an essential player in orchestrating inflammation and immune cell functions (*Baltimore, 2011*). However, the function of NF-κB in HSCs remains relatively unexplored. Under stress-free conditions, NF-κB is not essential for HSC function, because mice genetically deleted for NF-κB subunits, such as *Nfkb1* (also known as p50) and *Rel* (also known as c-Rel), have no apparent developmental abnormality in the hematopoietic system. Mice engrafted with p50 or c-Rel knockout HSCs or RelA knockout fetal HSCs also develop

**eLife digest** Hematopoietic stem cells are cells that both renew themselves and develop into any type of blood cell, including red blood cells and the several classes of immune cells. When an injury or infection occurs, it is vital that hematopoietic stem cells replenish themselves in addition to developing into the new blood cells that are needed to help the body recover. Injury and infection also lead to the inflammatory response: tissue becomes inflamed as cytokines and other molecules are released at the site of the damage to help maintain the body's immunity. It is thought that inflammatory molecules directly affect the rate at which stem cells become immune cells, with the protein NF-κB having an important role, but the details of this process are not fully understood.

To explore the connections between hematopoietic stem cells and the inflammatory response, Zhao et al. bred mice that do not produce a type of RNA called microRNA-146a. In wild-type mice, this RNA would inhibit the production of NF-κB, so the mutant mice have abnormally high levels of NF-κB. They found that the rate at which stem cells were being converted into immune cells in the mutant mice was so high that the stores of stems cells became exhausted, which was very detrimental to the health of the mice. They also went on to identify the signaling pathways that microRNA-146a influences in order to maintain supplies of stem cells and an adequate inflammatory response in healthy mice.

Zhao et al. also studied individuals with human myelodysplastic syndrome, a severe blood disorder that is associated with faulty hematopoietic stem cells, and found that these individuals produce relatively little microRNA-146a. The establishment of a link between microRNA-146a and having an adequate level of hematopoietic stem cells could have implications for human health, given the importance of these cells in both the aging process and the immune response.

relatively normal immune system under stress-free conditions (*Gerondakis et al., 2012*). However, mice with activated NF-κB signaling, as a consequence of deleting IκBα, A20, or the inhibitory domain of p50 or p52 subunits of NF-κB, display severe inflammation, early lethality, and complex phenotypes, making studies of HSCs difficult to perform and interpret (*Lee et al., 2000*; *Gerondakis et al., 2006*).

In recent years, microRNAs (miRNAs) have emerged as a class of small noncoding RNAs involved in the regulation of NF-κB (*Boldin and Baltimore, 2012*). Among them, miR-146a has been shown to be a particularly important negative regulator of NF-κB by targeting two upstream signal transducers, TRAF6 and IRAK1. Mice with targeted miR-146a deletion represent one of the first genetic mouse models with NF-κB-driven chronic and low-grade inflammation that develops spontaneously with aging and can be accelerated by repeated stimulation, allowing investigation of the long-term effects of chronic inflammation and NF-κB activation on HSCs and oncogenic processes (*Boldin et al., 2011*; *Zhao et al., 2011*).

Given this background, we have used miR-146a-deficient mice to examine the function of miR-146a and NF-κB in HSCs and progenitor cells during chronic inflammation and to directly test a long-standing hypothesis that chronic inflammation promotes excessive HSC and progenitor cell proliferation and differentiation and can lead to eventual HSC exhaustion and pathological myelopoiesis. Here, we demonstrate that this single miRNA, miR-146a, functions as a critical guardian of HSC quality and longevity during chronic inflammatory stress in mice. In the absence of miR-146a, HSC homeostasis is disrupted under physiological stresses such as aging and periodic bacterial encounters, as indicated by declines of HSC number and quality and dysregulated HSPC proliferation and differentiation. Chronically, these nominal stressors can lead to severe pathologies, such as HSC exhaustion, bone marrow failure, and myeloproliferative disease, produced by chronic NF-κB hyperactivation and IL-6 overproduction. This study speaks to a molecular pathway involving miR-146a/TRAF6/NF-κB/IL-6 that links chronic inflammatory stresses to the functional decline and depletion of HSCs and the development of myeloproliferative diseases.

## Results

### MiR-146a regulates HSC numbers during chronic inflammatory stress

To examine the role of miR-146a, we first assessed the expression of miR-146a and its related family member, miR-146b, during hematopoietic differentiation. We purified by FACS various types of

hematopoietic stem and progenitor cell (HSPC) populations from young wild-type (WT) mice. We found that miR-146a and miR-146b were expressed at variable levels throughout hematopoietic development. The expression of miR-146a increased by twofold as long-term HSCs (defined as Lineage⁻Sca1⁺cKit⁺ CD150⁺CD48⁻) differentiated into a mixed pool of short-term HSCs and multipotent progenitor cells (MPPs) (defined by Lineage⁻Sca1⁺cKit⁺, referred to as LSK cells). The lowest expression of miR-146a was detected in myeloid progenitor cells (defined by Lineage⁻Sca1⁻cKit⁺, referred to as L⁻S⁻K⁺ cells) (*Figure 1A*). In comparison, miR-146b expression was more uniform throughout hematopoietic development (*Figure 1A*). This expression pattern suggests that miR-146a and miR-146b could be functional in cells as primitive as the long-term HSCs and throughout hematopoietic development.

To characterize the physiological function of miR-146a in HSCs, we examined the consequences of miR-146a deficiency on various hematopoietic cells using mice with a targeted deletion of the *Mir146a* gene (*Boldin et al., 2011*; *Zhao et al., 2011*). We found identical numbers of phenotypically defined subsets and equal colony-forming ability in vitro in 6-week-old WT and *Mir146a⁻/⁻* (miR-146a KO) mice (*Figure 1—figure supplement 1A–C*). These data indicate that deleting miR-146a has no detectable effect on hematopoiesis early on in life in a standard pathogen-free environment.

We have previously shown that miR-146a KO mice develop spontaneous inflammation as they age (*Boldin et al., 2011*; *Zhao et al., 2011*). To characterize the role of miR-146a in HSCs during low level of chronic inflammation, we allowed age- and sex-matched WT and miR-146a KO mice to age over a year. By 4 months, miR-146a KO mice developed a mildly hypercellular bone marrow indicated by an increase in total bone marrow CD45⁺ cells, LSK cells, and long-term HSCs, with unaltered percentages of CD19⁺, CD11b⁺ and CD3ε⁺ cells (*Figure 1—figure supplement 1D*). However, the increase in bone marrow HSPCs and mature cells was not sustained. By 8 months of age, miR-146a KO mice showed a significant decrease in the number of total bone marrow cells and phenotypically defined HSPCs, including LSK cells and CD150⁺CD48⁻ or EPCR⁺ long-term HSCs (*Figure 1B,C*). The depletion became progressively more severe by 12 months of age when the majority of miR-146a KO mice showed only a residual number of CD45⁺ bone marrow cells and nearly complete exhaustion of HSCs (*Figure 1D*). To understand the cellular process leading to a transient hypercellular marrow and eventual HSC exhaustion, we analyzed the bone marrow and spleen of aging miR-146a KO mice. Dysregulated hematopoiesis was observed in 8-month-old KO mice, as shown by an increased percent of LSK cells within the total bone marrow but a significantly decreased fraction of long-term HSCs among the LSK cells (*Figure 1E*). By this age, miR-146a KO mice have already developed a prominent myeloproliferative phenotype (*Zhao et al., 2011*). In addition, miR-146a KO mice also showed a significant increase in the number of LSK cells and long-term HSCs in their spleens, which may be due to HSPC mobilization from bone marrow to spleen and/or de novo splenic HSPC proliferation in response to bone marrow failure (*Figure 1F*). Furthermore, when total HSCs from the marrow and spleen of the same mice were summed, a reduction in HSC number was still apparent (*Figure 1G*). Thus, although miR-146a KO mice contain normal levels of HSPCs when they are young, as they age, they go through a hypercellular stage and then eventually start to lose bone marrow HSCs and differentiated cells, leading to HSC exhaustion and bone marrow failure. This suggests to us that a dysregulated HSC differentiation toward the myeloid lineage is taking place in the marrow, accompanied by an increased appearance of HSCs in the spleen, probably as a form of homeostatic compensation.

Importantly, we found that chronic inflammatory stimulation with bacterial components in young miR-146a KO mice was sufficient to accelerate the development of the same hematopoietic defects seen during aging of these mice. We stimulated 8-week-old WT and miR-146a KO mice with LPS repeatedly for a month and observed a dysregulated HSC differentiation toward myeloid cells in miR-146a KO mice, compared to WT mice, a phenotype similar to the one that spontaneously occurs during aging of the KO mice (*Figure 1H–J* and *Figure 1—figure supplement 1E*). Thus, miR-146a is needed to maintain HSC homeostasis in response to chronic inflammation. This suggests that the stress of chronic inflammation may be the physiologically relevant stimulator of HSC deficiency and myeloproliferative disease in the miR-146a KO mice. Perhaps, this becomes evident in aging miR-146a KO mice not subjected to experimental inflammatory stimulation because of a low-level continual exposure to bacterial materials even in our relatively clean conditions of animal husbandry. Whether true pathogens or commensals might be the inducing agents will require further investigation.

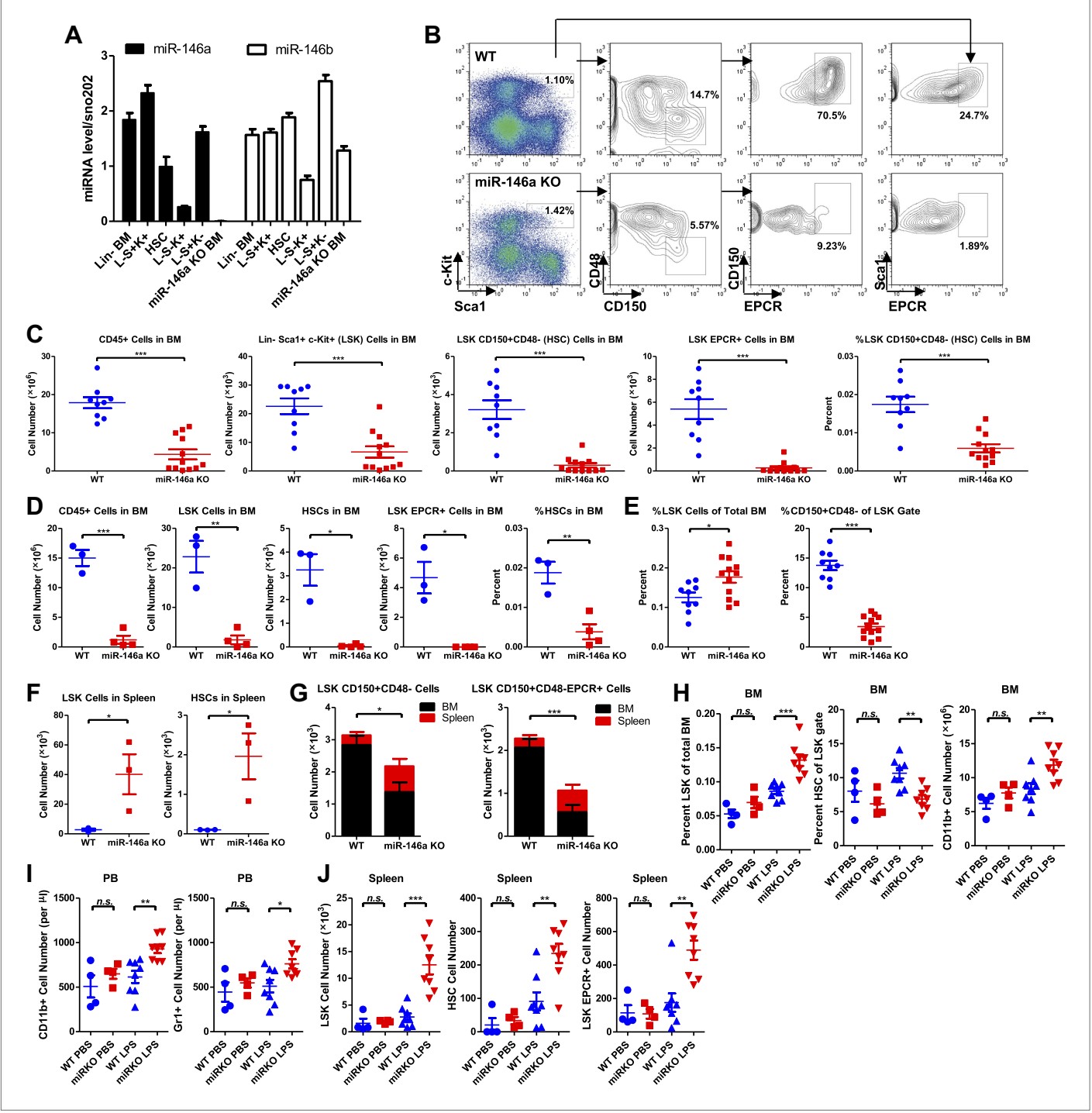

**Figure 1**. Accelerated HSC decline and myeloproliferation in miR-146a–deficient mice during chronic inflammation. (**A**) MiR-146a and miR-146b expression in FACS-sorted HSPC populations by Taqman RT-qPCR. Lin-BM, lineage negative bone marrow cells; L⁻S⁺K⁺ (LSK), Lin⁻Sca1⁺cKit⁺; HSC, LSK CD150⁺CD48⁻; L⁻S⁻K⁺, Lin⁻Sca1⁻cKit⁺; L⁻S⁺K⁻, Lin⁻Sca1⁺cKit⁻; miR-146a KO BM, total bone marrow cells from *Mir146a⁻/⁻* mice. (**B**). Representative FACS plots of LSK cells and CD150⁺CD48⁻ or EPCR⁺ HSCs from BM of 8-month-old wild-type (WT) and miR-146a KO mice. Quantification of number of BM CD45⁺ cells, LSK cells, LSK CD150⁺CD48⁻ HSCs, LSK EPCR⁺ HSCs and percent of LSK CD150⁺CD48⁻ HSCs of total BM from 8-month-old (**C**) and 12-month-old (**D**) WT and miR-146a KO mice by FACS. (**E**) Quantification of percent of LSK cells of total BM and percent of HSCs of LSK gate from BM of 8-month-old WT and miR-146a KO mice by FACS. (**F**) Quantification of number of LSK cells and LSK CD150⁺CD48⁻ HSCs from spleen of 8-month-old WT and miR-146a KO mice by FACS. (**G**). Total number of LSK CD150⁺CD48⁻ or LSK CD150⁺CD48⁻ HSCs from BM and spleen of 6-month-old WT and miR-146a KO mice. (**H**)–(**J**) 8-Week-old WT and miR-146a KO (miR KO) mice were subjected to repeated low-dose of intraperitoneal LPS stimulation

*Figure 1. Continued on next page*

Figure 1. Continued

(1 mg LPS/kg of body weight for 8 times) or PBS control spread over a month. At the end the month, four groups of mice were harvested for FACS analysis. (H) Quantification of percent of LSK cells of total BM, CD150+CD48− HSCs of LSK gate, and number of CD11b+ myeloid cells in BM. I. Number of myeloid cells (CD11b+ or Gr1+) in peripheral blood. (J) Quantification of total number of LSK CD150+CD48− HSCs, LSK EPCR+ HSCs and LSK cells in spleen.

The following figure supplements are available for figure 1:

Figure supplement 1. HSPC FACS analysis and colony-forming ability in 6-week-old and 4-month-old WT and miR-146a KO mice.

## MiR-146a-deficiency results in progressive decline in the quality of HSCs

In addition to the progressive loss of phenotypically defined HSCs in aging miR-146a KO bone marrow, we found that the functional quality of HSCs deteriorates in mice lacking miR-146a. To assess HSC function, we compared WT and miR-146a KO bone marrow HSCs in their ability to generate the entire hematopoietic repertoire competitively in vivo. Total bone marrow cells from either 6-week-old WT or miR-146a KO mice, both which were CD45.2+, were transplanted along with an equal number of CD45.1+ WT bone marrow cells, into lethally irradiated CD45.1+ WT recipient mice (*Figure 2A*). 6 months after transplant, CD45.2+ and CD45.1+ cells in both the CD45.2 WT/CD45.1 WT and CD45.2 KO/CD45.1 WT mice contributed identical proportions of cells in nearly all mature hematopoietic lineages and HSPCs (*Figure 2B*). Furthermore, when we purified long-term HSCs (defined by LSK CD150+CD48−) from WT or miR-146a KO bone marrow for competitive repopulation assay, we again observed a similar contribution of WT or miR-146a KO HSCs to total white blood cells and HSPCs 6 months after transplantation (*Figure 2—figure supplement 1A*). However, when we extended the experiment past 10 months, we began to observe a decreased contribution of miR-146a KO cells to long-term HSCs and LSK cells, but no reduction in CD45+ cells, in the bone marrow of CD45.2 KO/CD45.1 WT mice (*Figure 2C,D*). Interestingly, while WT/WT chimera mice had similar numbers of CD45.2+ and CD45.1+ HSCs, KO/WT chimera mice showed a specific reduction in only the CD45.2+ KO HSCs and a slight elevation of CD45.1+ WT HSCs, possibly as a compensation for the loss of KO HSCs (*Figure 2E*). These data indicate that miR-146a-deficient long-term HSCs from young mice have an intrinsic defect and are more susceptible to depletion compared to WT HSCs in the same environment; however, the intrinsic defect of miR-146a KO HSCs from young mice is rather modest and takes 10 months to become apparent in this transplant setting.

To investigate how aging affects the quality of miR-146a KO HSCs, we used 4-month-old, instead of 6-week-old, miR-146a KO and WT mice for competitive repopulation assay. At 4 months of age, miR-146a KO mice display a modestly hypercellular bone marrow and mildly increased phenotypically defined HSCs, but no disease phenotype (*Figure 1—figure supplement 1D*). Surprisingly, miR-146a KO cells were outcompeted by their WT counterparts as early as the first month after the transplant, and there was a steady decline in their percentage over a period of 6 months (*Figure 2F–H*). 6 months after transplant, neither group of mice showed signs of pathology, and they had identical levels of total mature hematopoietic cells and HSPCs (*Figure 2—figure supplement 1B–E*). However, when comparing the contribution of CD45.1+ cells vs CD45.2+ cells to the total pool, we observed a significant disadvantage of miR-146a KO HSPCs and mature lineages (*Figure 2I*). This is consistent with the preferential loss of CD45.2+ KO LSK cells and HSCs, but not the cotransplanted CD45.1+ WT cells, in KO/WT chimera mice (*Figure 2J,K*). Similar to the progressive decline of HSC number, the functional quality of HSCs became more and more compromised as miR-146a KO mice aged. When the competitive repopulation experiment was carried out with 6-month-old WT and miR-146a KO bone marrow cells, at which time increased myeloid and LSK cells and decreased long-term HSCs were already evident (*Figure 2—figure supplement 1F*), we observed that miR-146a KO cells were completely overwhelmed by their WT counterparts in the recipient mice, with ratios of about 1 miR-146a KO cell to 10 WT cells for nearly all mature lineages and HSPCs (*Figure 2—figure supplement 1G*). The defective repopulating ability observed in all mature and HSPC lineages, including the long-term HSCs, in all three major hematopoietic compartments strongly suggests that the defect must originate in the most primitive HSCs. These data indicate that miR-146a-deficiency has a detrimental effect on the quality of HSCs under chronic inflammatory stress. The functional decline of HSCs in the competitive repopulation

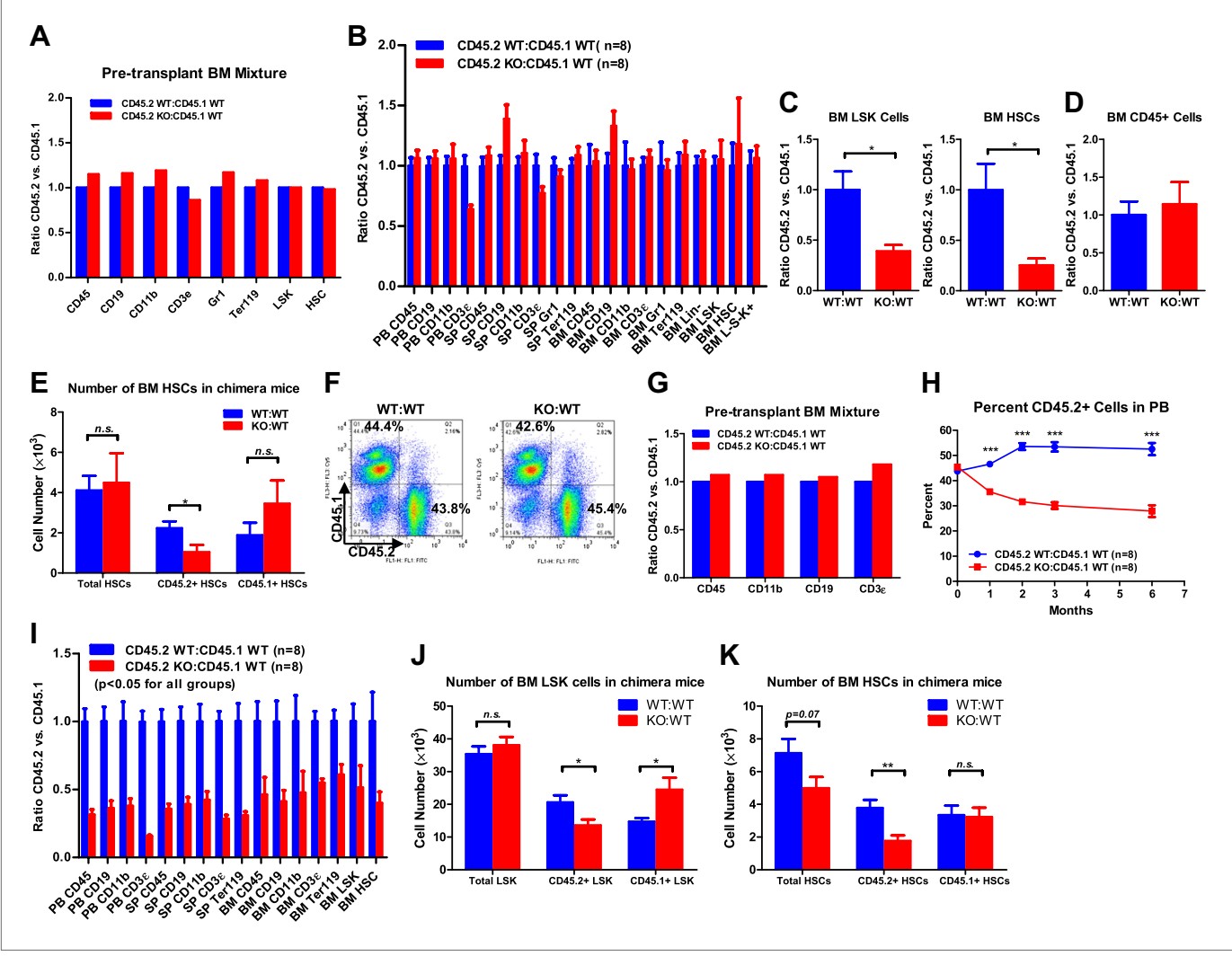

**Figure 2**. Progressive functional decline of miR-146a-deficient HSCs. (**A**) Ratio of CD45.2$^+$ over CD45.1$^+$ cells of various lineages in pretransplanted bone marrow (BM) mixtures consisted of equal numbers of CD45.2$^+$ WT and CD45.1$^+$ WT total BM cells (blue bar, CD45.2 WT:CD45.1 WT) or equal numbers of CD45.2$^+$ miR-146a KO and CD45.1$^+$ WT total BM cells (red bar, CD45.2 KO:CD45.1 WT). FACS analysis was performed on the BM mixtures before transplantation to determine the starting ratios of various lineages. (**B**) Ratio of CD45.2$^+$ over CD45.1$^+$ cells of various lineages in peripheral blood (PB), spleen (SP) and BM 6 months after transplantation. Blue bar, CD45.2 WT:CD45.1 WT, represents mice received CD45.2 WT:CD45.1 WT BM cells; red bar, CD45.2 KO:CD45.1 WT, represents mice received CD45.2 KO:CD45.1 WT BM cells. All donor mice were 6-week-old female and recipient mice were 2-month-old CD45.1$^+$ WT female. Ratio of CD45.2$^+$ over CD45.1$^+$ of BM LSK cells and HSCs (**C**) and total white blood (CD45$^+$) cells (**D**) 10-month after transplantation. (**E**) Number of total HSCs, CD45.2$^+$ HSCs and CD45.1$^±$ HSCs in recipient chimera mice. WT:WT, CD45.2 WT:CD45.1 WT; KO:WT, CD45.2 KO:CD45.1 WT. n = 8 for each group. (**F**)–(**K**) A repeat of the above experiment with age-and-sex-matched 4-month-old WT and miR-146a KO female mice. (**F**) Representative FACS plots of BM mixtures before transplantation showing CD45.2/CD45.1 ratio close to 1 for both WT:WT and KO:WT BM mixtures. (**G**) Ratio of CD45.2$^+$ over CD45.1$^+$ cells of various lineages before transplantation. (**H**) Percentage of CD45.2$^+$ cells of CD45$^+$ peripheral blood nucleated cells at 1, 2, 3 and 6 months. (**I**) Ratio of CD45.2$^+$ over CD45.1$^+$ cells of various lineages in PB, SP, and BM 6 months after transplantation. All ratios of CD45.2 WT:CD45.1 WT are normalized to 1. Number of total, CD45.2$^+$ and CD45.1$^+$ LSK cells (**J**) and HSCs (**K**) in recipient chimera mice. LSK, Lin-cKit$^+$Sca1$^+$; HSC, LSK CD150$^+$CD48$^-$; L$^-$S$^-$K$^+$, Lin$^-$Sca1$^-$cKit$^+$.

The following figure supplements are available for figure 2:

**Figure supplement 1**. Functional decline of miR-146a-deficient HSCs.

setting is evident in healthy 8-week-old miR-146a KO mice and becomes significant in 4-month-old mice in the absence of any observable pathology, indicating the physiological importance of miR-146a as a guardian of HSC quality and longevity.

## Hematopoietic-intrinsic and hematopoietic-extrinsic contributions to HSC defects and pathological hematopoiesis

To directly examine both hematopoietic-intrinsic defects and extrinsic factors on hematopoiesis in the absence of miR-146a, we performed reciprocal bone marrow transplants, transferring WT bone marrow cells into miR-146a KO recipient mice (WT to KO) and KO bone marrow cells into WT recipient mice (KO to WT). WT to WT and KO to KO transplant mice were included as controls. After 5 months, we harvested all groups for analysis. Interestingly, mice of the WT to WT and WT to KO groups had identical levels of HSPCs in their bone marrows and spleens, suggesting that the miR-146a-deficient environment is not sufficient to induce significant HSC abnormalities in WT cells during this time (*Figure 3A–G*). In comparison, mice in the KO to WT and KO to KO groups both showed twofold to threefold reductions in bone marrow HSCs and myeloid progenitor cells, but not in LSK cells (*Figure 3A–C*), and about a 10-fold increase in spleen HSPCs (*Figure 3D–G*). As in the miR-146a germline KO mice, an increased percentage of LSK cells and a decreased representation of long-term HSCs within the LSK fraction was also observed in these transplantation groups, indicating a dysregulated HSC homeostasis (*Figure 3H–J*). In addition to the HSPC abnormality, mice of the KO to WT and KO to KO groups also developed pathological features recapitulating those seen in aged miR-146a germline KO mice (*Boldin et al., 2011*; *Zhao et al., 2011*). By 5 months post-transplant, two out of eight mice in the KO to KO transplant group had succumbed to tumor pathology, including one case of CD4+ T-cell lymphoma in the thymus and one case of kidney tumor, with no tumors observed in any of the other groups (*Figure 3—figure supplement 1A,B*). Necropsy also showed splenomegaly and pale bone marrows in the KO to WT and the KO to KO groups (*Figure 3—figure supplement 1C,D*). Myeloproliferation was a prominent feature in mice of the KO to WT and the KO to KO groups, which showed an increase of spleen weight, number of white blood cells, B cells, T cells, and, most dramatically, number of CD11b+ or Gr1+ myeloid cells in their spleens, compared to the WT to WT and the WT to KO groups (*Figure 3—figure supplement 1E–L*). Increased myeloproliferation/myelopoiesis was also observed in the bone marrow and peripheral blood (*Figure 3—figure supplement 1M,N*). Moreover, in line with HSC exhaustion and bone marrow failure, mice of the KO to WT and KO to KO groups exhibited hypocellular bone marrow and peripheral cytopenia (*Figure 3—figure supplement 1O–U*).

These data indicate that it is the miR-146a deficiency in hematopoietic cells that plays the dominant role in determining the phenotype of the miR-146a knockout mouse, because transferring miR-146a-deficient bone marrow cells into WT environment, but not the reciprocal transfer, is sufficient to yield a majority of the phenotypes seen in mice with miR-146a deleted in both hematopoietic and nonhematopoietic cells. However, the miR-146a-deficient environment has a contributory role to the overall enhanced myelopoiesis because WT bone marrow cells in the KO environment gave rise to a mild but statistically significant increase in the number of CD45+ and CD11b+ marrow cells and the percentage of CD11b+ and Gr1+ cells in their spleens, compared to the WT to WT mice (*Figure 3—figure supplement 1K,L,O,P*). More indicative is that when WT bone marrow cells that were transiently transplanted into miR-146a KO environment for 2 months were competed against WT control bone marrow cells (*Figure 3K*), they showed a modest advantage in generating myeloid and lymphoid lineages in their spleens, peripheral blood and to a lesser extent bone marrows, whereas bone marrow HSPCs remained unaffected (*Figure 3L–O*). Overall, we have shown that hematopoietic-intrinsic deficiency of miR-146a plays the dominant role in driving the HSC defects and pathological myeloproliferation while the miR-146a deficient environment also contributes to the overall phenotype. In the absence of the driving force from the miR-146a deficient environment, lethal tumor pathology and HSC functional decline are attenuated or delayed.

## Lymphocytes contribute to the progressive loss of HSCs and myeloproliferative disease

We have shown that hematopoietic-intrinsic factors play the predominant role in the dysregulated hematopoiesis of miR-146a-deficient mice. But this does not tell us which cell type(s) in the hematopoietic compartment might influence aspects of hematopoiesis. Because miR-146a–deficient lymphocytes display a hyperactivated phenotype with dysregulated cytokine production (*Yang et al., 2012*), we

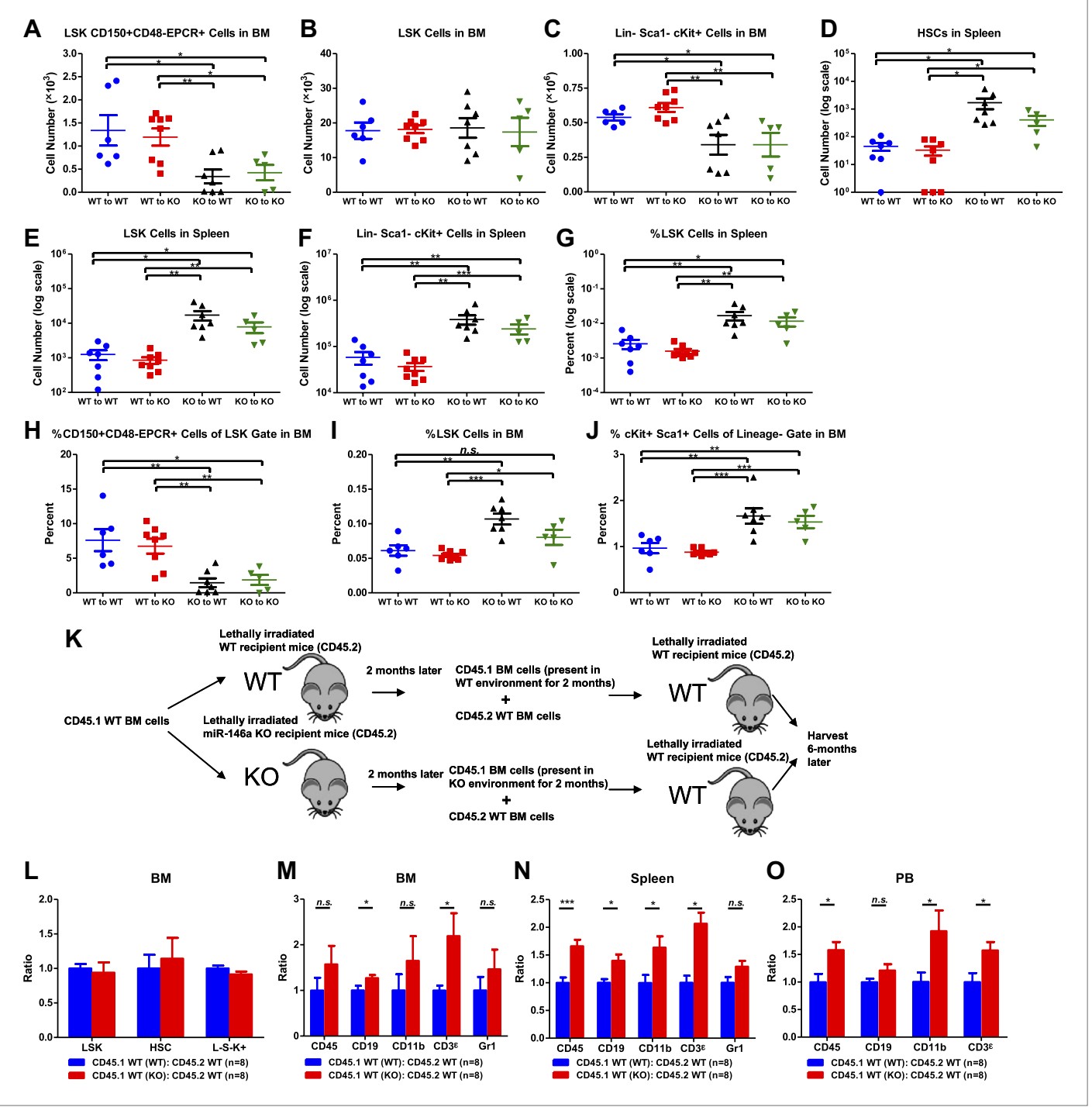

**Figure 3**. Hematopoietic-intrinsic and hematopoietic-extrinsic contribution to hematopoietic defects. (**A**)–(**J**) Reciprocal bone marrow (BM) transplant performed by transferring WT donor BM cells to WT recipient mice (WT to WT), WT donor BM cells to miR-146a KO recipient mice (WT to KO), miR-146a KO donor BM cells to WT recipient mice (KO to WT), and miR-146a KO donor BM cells to miR-146a KO recipient mice (KO to KO). All donor and recipient mice were 8-week-old female mice. Mice were harvested for analysis at the end of 5 months. Quantification of total number of HSPCs in spleen and BM, including LSK CD150+CD48−EPCR+ HSCs (**A**), LSK (Lin−cKit+Sca1+) cells (**B**), Lin−cKit+Sca1− cells (**C**) in BM, and LSK CD150+CD48− HSCs (**D**), LSK cells (**E**), and Lin−cKit+Sca1− cells (**F**) in spleen. Quantification of percent of HSPCs in spleen and BM, including percent LSK cells in spleen (**G**), percent of CD150+CD48−EPCR+ HSCs in LSK gate in BM (**H**), percent of LSK cells in total BM (**I**), and percent of cKit+Sca1+ in Lin− gate in BM (**J**). (**K**)–(**O**). Serial BM transplant performed by first transplanting CD45.1+ WT BM cells into either CD45.2+ WT or miR-146a KO recipient mice for 2 months, which were then harvested and mixed with CD45.2+ WT BM cells for second transplantation into CD45.2+ WT recipient mice. Mice received CD45.1 WT (WT):CD45.2 WT

*Figure 3. Continued on next page*

*Figure 3. Continued*

or CD45.1 WT (KO):CD45.2 WT cells were harvested 6 months later for FACS analysis. (**K**) Schematic diagram of the experimental setup. (**L**) Ratio of CD45.1$^+$ over CD45.2$^+$ cells of BM HSPCs, including LSK cells, LSK CD150$^+$CD48$^-$ HSCs and L$^-$K$^+$S$^-$ cells. Ratio of CD45.1$^+$ over CD45.2$^+$ cells of various lineages in BM (**M**), spleen (**N**), and PB (**O**), including CD45$^+$, CD19$^+$, CD11b$^+$, CD3ε$^+$ and Gr1$^+$ cells.

The following figure supplements are available for figure 3:

**Figure supplement 1**. Hematopoietic-intrinsic and hematopoietic-extrinsic contribution to hematopoietic defects.

first determined whether dysregulation of miR-146a-deficient lymphocytes might contribute to HSC depletion in miR-146a KO mice. To this end, we crossed mice with a targeted deletion of the *Rag1* gene, which is required for lymphocyte maturation, with miR-146a KO mice to generate *Mir146a$^{-/-}$ Rag1$^{-/-}$* double knockout mice (miR/Rag1 DKO). When WT, miR-146a KO (miR KO), Rag1 KO, and miR/Rag DKO mice were allowed to age for 10 months, the miR KO mice showed significant depletion of HSPCs (*Figure 4A–D*), a finding consistent with what we have observed previously in an independent examination of WT and miR-146a KO mice (*Figure 1*). Specifically, long-term HSCs and LSK cells in the miR KO marrow were reduced to only 3% and 15% of the respective WT levels. In comparison, depletion of HSPCs in miR/Rag1 DKO bone marrow was partially rescued. Long-term HSCs and LSK cells in miR/Rag1 DKO bone marrow rose to 30% and 50% of the Rag1 KO levels, respectively, and the numbers of myeloid progenitor cells and total white blood cells were normal (*Figure 4A–E*). These data indicate that miR-146a-deficient lymphocytes contribute substantially to the overall HSPC exhaustion. In addition, miR-146a-deficient lymphocytes were shown to be a major driver of the development of bone marrow failure and myeloproliferative disease, because miR/Rag1 DKO mice showed normal bone marrow cellularity and an attenuated splenomegaly and myeloproliferative phenotype (*Figure 4E–L*). This was also confirmed by histological analysis of femur bones of miR/Rag1 DKO mice, which showed absence of marrow fibrosis, a common and prominent feature of aged miR KO mice (*Figure 4H*). However, the rescue was not complete because miR/Rag1 DKO mice still had mildly enlarged spleens and a twofold to threefold increase in the number of total white blood cells and myeloid cells compared to Rag1 KO mice, indicating that miR-146a deficiency in myeloid lineages has an intrinsic effect on the development of myeloproliferation (*Figure 4M*). It is worth noting that the intrinsic effect of miR-146a deficiency in myeloid cells may either be in *cis* or *trans*: it could be the proliferating cells responding to a changed intracellular signaling or to factors secreted by myeloid cells acting in an autocrine or paracrine manner.

## NF-κB regulates HSC homeostasis under chronic inflammatory stress

To understand the molecular mechanism responsible for the stress-induced hematopoietic dysregulation, we focused on the main pathway known to be regulated by miR-146a, the NF-κB pathway. We have previously shown that aging miR-146a KO mice display hyperactivated NF-κB activity that is responsible for the development of myeloid malignancy (*Zhao et al., 2011*). To study whether NF-κB may also regulate HSC homeostasis under inflammatory stress, we used a transgenic NF-κB–GFP reporter mouse to monitor NF-κB activity quantitatively and efficiently in various cell types by measuring GFP fluorescence (*Magness et al., 2004*; *Lippert et al., 2009*). We first tested whether NF-κB can be activated in LSK cells and HSCs in 2-month-old WT NF-κB-GFP (WT-GFP) reporter mouse under steady state and after LPS stimulation. We found that about 6–7% of WT LSK cells and HSCs had basally activated NF-κB activity, as measured by GFP expression. 6 hr after LPS challenge in vivo, the percentage of LSK cells and HSCs with activated NF-κB increased to more than 25% (*Figure 5—figure supplement 1A*). Interestingly, 8-month-old WT-GFP mice also showed an increased percentage of LSK cells and HSCs with basal NF-κB activation, compared to 2-month-old mice (*Figure 5—figure supplement 1A*). These data suggest that both LSK cells and HSCs have functional NF-κB–mediated transcription that can be augmented by LPS stimulation and aging. To analyze whether hyperactivated NF-κB activity is a feature in miR-146a KO HSPCs, we bred NF-κB-GFP reporter mice with miR-146a KO mice to generate *Mir146a$^{+/+}$* (WT-GFP), *Mir146a$^{+/-}$* (miRHET-GFP), and *Mir146a$^{-/-}$* (miRKO-GFP) NF-κB-GFP reporter mice. Unperturbed 8-week-old mice of WT-GFP, miRHET-GFP, and miRKO-GFP genotypes showed identical levels of basal NF-κB activity (*Figure 5—figure supplement 1B–D*). After repeated LPS stimulation, miRKO-GFP mice, in comparison to WT-GFP and miRHET-GFP mice, showed increased percentages of GFP$^+$ cells in various HSPCs (*Figure 5A–C*) and mature cells in their bone

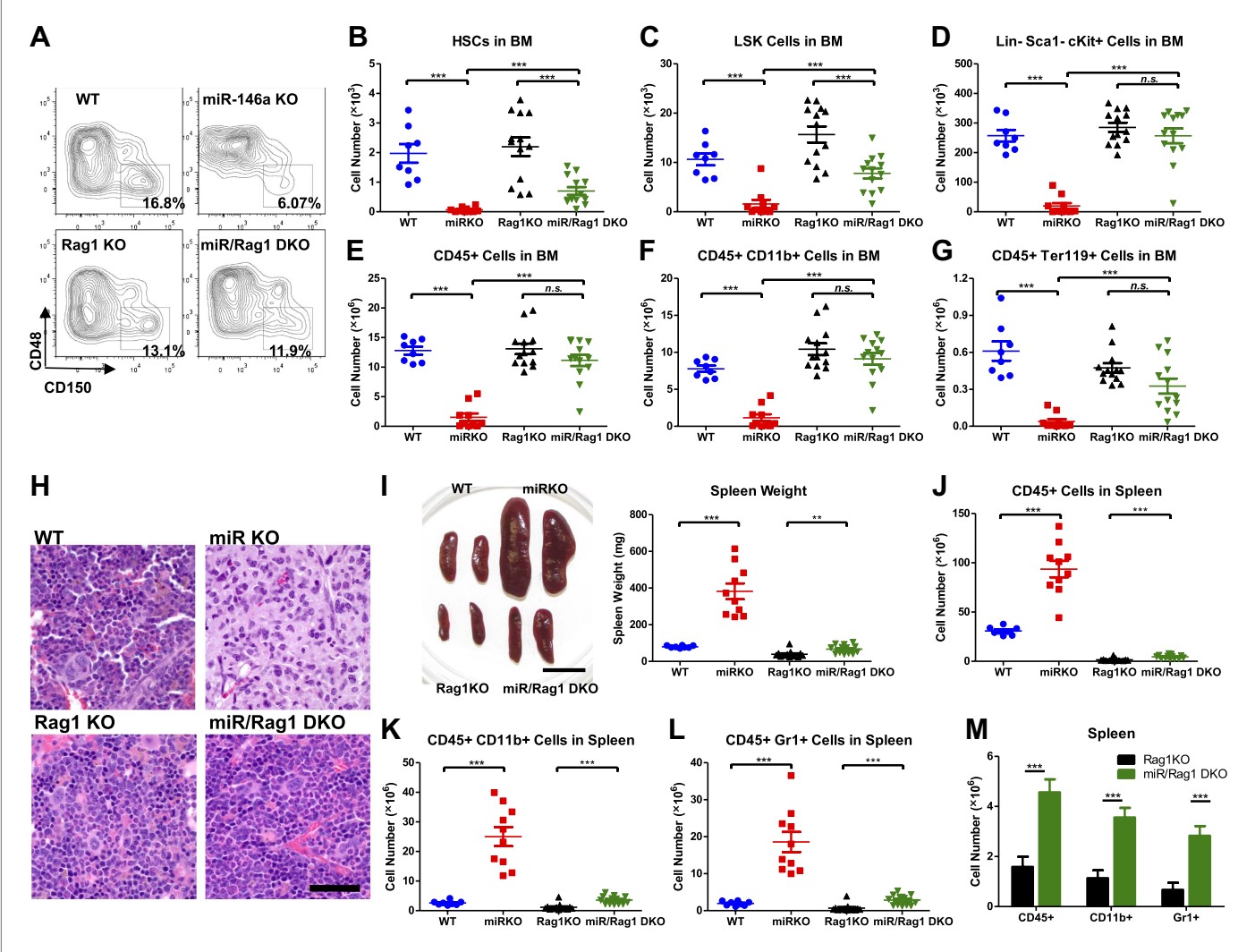

**Figure 4**. MiR-146a-deficient lymphocytes contribute to the HSC defect and myeloproliferation. (**A**)–(**M**) Age-and-sex-matched WT, miR-146a KO (miR KO), Rag1 KO and miR-146a/Rag1 double KO (miR/Rag1 DKO) mice were allowed to age to 10-month-old before harvested for analysis. (**A**) Representative FACS plots of CD150+CD48– HSCs of the LSK gate. Quantification of total number of HSCs (**B**), LSK cells (**C**), and Lin–cKit+Sca1– myeloid progenitor cells (**D**) in BM. Quantification of total number of CD45+ (**E**), CD11b+ (**F**), and Ter119+ (**G**) cells in BM. (**H**) Representative histological pictures (H&E stain) of femur bones. Scale bar, 40 μm. (**I**). Representative photograph of spleens and spleen weight. Total number of CD45+ (**J**), CD11b+ (**K**), and Gr1+ (**L**) cells in spleen. (**M**) For comparison, various cell lineages in spleen of only Rag1 KO and miR/Rag1 DKO mice were regraphed from (**J–L**).

marrows, spleens and peripheral blood (*Figure 5—figure supplement 2A–C*), demonstrating that chronic inflammatory stimulation with bacterial components leads to hyperactivated NF-κB activity in young miR-146a-deficient mice.

To further investigate whether the increased NF-κB activity is responsible for driving HSC depletion in bone marrow, we deleted a main subunit of NF-κB, p50, to determine whether reduced NF-κB activity in miR-146a/p50 double-knockout (miR/p50 DKO) mice might rescue the HSC exhaustion. By 8–9 months of age, miR/p50 DKO mice still exhibited levels of white blood cells and long-term HSCs comparable to that of WT mice, indicating a rescue of the HSC defects (*Figure 5D*). We have previously shown that p50 deletion rescues the myeloproliferative disease in 6-month-old miR-146a KO mice (*Zhao et al., 2011*). To determine whether myeloid cancer can also be reversed in miR/p50 DKO mice, we aged a cohort of WT, miR KO, p50 KO, and miR/p50 DKO mice to about 1½ years, by which time about 50% of miR-146a KO mice will have developed

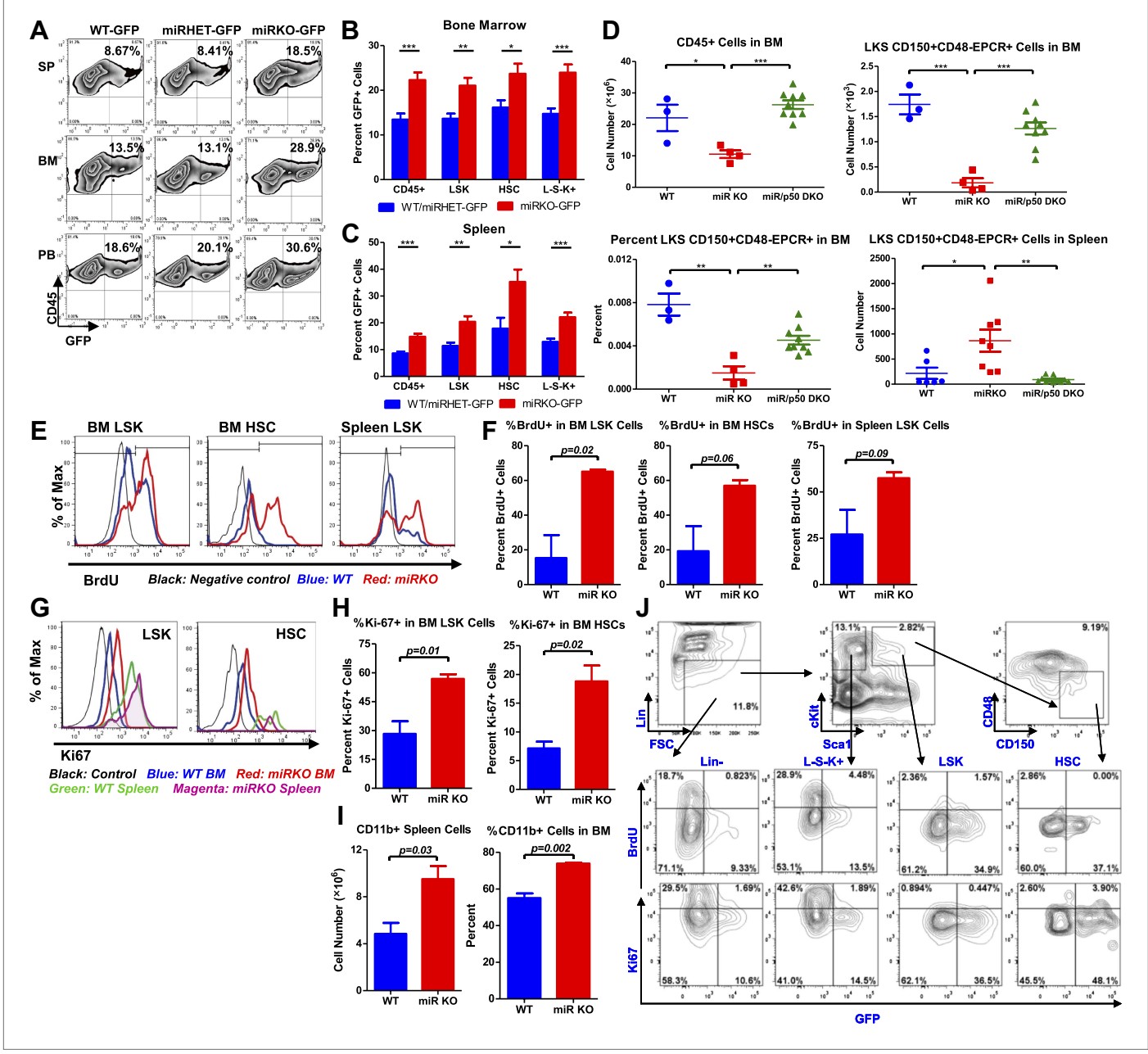

**Figure 5**. NF-κB regulates HSC homeostasis during chronic inflammation. (**A**)–(**C**) 8-Week-old WT (WT-GFP), *Mir146a+/−* (miRHET-GFP), and *Mir146a−/−* (miRKO-GFP) NF-κB-GFP reporter mice were subjected to repeated intraperitoneal LPS stimulation (3 mg LPS/kg of body weight every other day) for 1 week. Percent of GFP+ cells in various lineages were quantified by FACS. (**A**) Representative FACS plots of GFP+ white blood cells (CD45+) in spleen (SP), bone marrow (BM), and peripheral blood (PB). Quantification of percent GFP+ cells of HSPCs in bone marrow (**B**) and spleen (**C**), including CD45+, LSK cells, HSCs (LSK CD150+CD48−) and L−S−K+ myeloid progenitor cells. (**D**) Age- and sex-matched WT, miR-146a KO (miR KO), and miR-146a/p50 double knockout (miR/p50 DKO) mice were allowed to age to 8–9 months before harvested for analysis. Quantification of total number of CD45+, LSK CD150+CD48−EPCR+ HSCs in BM, percent of LSK CD150+CD48−EPCR+ HSCs of total BM, and total number of LSK CD150+CD48−EPCR+ HSCs in spleen by FACS. (**E**)–(**I**) 8-Week-old WT and miR-146a KO (miR KO) mice were subjected to repeated low-dose of intraperitoneal LPS stimulation (1 mg LPS/kg of body weight) daily for 3 days. 1 mg of BrdU was injected intraperitoneally daily. BrdU+ and Ki-67+ HSPCs were quantified by FACS. (**E**) Representative FACS histograms of BrdU+ LSK cells and HSCs in BM and BrdU+ LSK cells in spleen. Blue: WT mice; red: miR KO mice. (**F**) Quantification of percent of BrdU+ cells in BM LSK cells, HSCs, and spleen LSK cells. (**G**) Representative FACS histograms of Ki-67+ LSK and HSCs in BM and spleen. Black: negative control; blue: WT BM; red: miR KO BM; green: WT spleen; magenta: miR KO spleen. (**H**) Quantification of Ki-67+ cells in BM LSK cells and HSCs. (**I**) Quantification of number and percent of CD11b+ myeloid cells in
*Figure 5. Continued on next page*

*Figure 5. Continued*

spleen and BM. (**J**) Representative FACS plots of BrdU⁺ or Ki-67⁺ and GFP⁺ cells of Lin⁻, L⁻S⁻K⁺, LSK and HSC in BM of 8-week-old WT-GFP mice stimulated with LPS (one dose, 1 mg/kg of body weight) for 4 hr.
The following figure supplements are available for figure 5:

**Figure supplement 1**. NF-κB activity in HSPCs and peripheral blood of NF-κB-GFP reporter mice.

**Figure supplement 2**. NF-κB regulates HSPC homeostasis during chronic inflammation.

myeloid cancer. Interestingly, miR/p50 DKO mice still showed reduced spleen weight and prolonged survival (*Figure 5—figure supplement 2D,E*). The incidence of tumors was also significantly reduced in miR/p50 DKO mice, compared to miR KO mice (*Figure 5—figure supplement 2F*). However, there were still two cases of splenic myeloid tumors from a total of 46 miR/p50 DKO mice analyzed. Histological analysis of spleens revealed that the majority of miR/p50 DKO spleens, in contrast to the miR KO spleens with frequent myeloid sarcoma, showed preserved lymphoid follicular structures (*Figure 5—figure supplement 2G*). Furthermore, significant marrow fibrosis seen in aged miR KO mice was also reversed in miR/p50 DKO mice (*Figure 5—figure supplement 2H*). Transplantation of miR/p50 DKO spleen cells into *Rag2⁻/⁻ Il2rg⁻/⁻* mice did not result in splenomegaly or myeloid pathology as transplanting miR KO splenic tumor cells did (*Figure 5—figure supplement 2I* and *Zhao et al., 2011*). Together, these data show that p50-deficiency significantly ameliorates the myeloproliferative disease/myeloid cancer and marrow fibrosis in aging miR-146a KO mice. These findings highlight the importance of the NF-κB pathway, particularly the p50 subunit, as the primary mediator of miR-146a deficiency driven myeloid oncogenesis and marrow failure. However, the modest but significant increase in spleen weight and the occasional occurrence of myeloid tumors and marrow fibrosis in aging miR/p50 DKO mice suggest that other NF-κB subunits or other pathways regulated by miR-146a may also mediate the disease phenotype. Overall, we have shown that chronic inflammation-induced or aging-associated NF-κB activation is responsible for driving HSC exhaustion, myeloproliferative disease, and myeloid cancer.

To explore whether the increased proliferation and cycling of miR-146a-deficient HSCs is an underlying cause of accelerated HSC depletion under chronic inflammation, we measured BrdU incorporation and Ki-67 expression in HSCs after LPS stimulation. Surprisingly, when examining 2-month-old WT and miR-146a KO mice, we did not observe a significant difference in the percentage of BrdU⁺ bone marrow LSK cells and HSCs whether the mice were unperturbed or stimulated with a single LPS injection for 12 hr (*Figure 5—figure supplement 2J*). However, after repeated LPS stimulation for 3 days, there were significantly higher percentages of BrdU⁺ and Ki-67⁺ miR-146a KO LSK cells and HSCs, compared to the WT cells, indicating an increased proliferation (*Figure 5E–H*). Consistent with increased HSC cycling and myeloid differentiation, miR-146a KO mice started to show a small but statistically significant increase in CD11b⁺ myeloid cells in their spleens and bone marrows after 3 days of LPS stimulation (*Figure 5I*). It is also interesting to note that identically gated LSK cells and HSCs in spleen seemed to be less quiescent than their bone marrow counterparts, as indicated by a higher level of Ki-67 expression after LPS stimulation, suggesting that the bone marrow milieu may be a better environment for maintaining HSC quiescence during inflammation (*Figure 5G*). These data show that chronic inflammation induces increased HSC proliferation and cycling in miR-146a KO mice and underscores the particular importance of miR-146a in modulating HSC activity during chronic inflammation, as opposed to in an acute setting.

Because we have shown that NF-κB is activated in many other hematopoietic cells, in addition to HSCs, we next determined whether NF-κB directly regulates HSC proliferation in a cell-autonomous manner or indirectly through cytokines produced by other cells. To this end, we injected WT-GFP mice with LPS and BrdU for 4 hr and measured both NF-κB activation and proliferation simultaneously. Interestingly, we only found a small fraction of cells that were doubly positive for GFP and BrdU or Ki67, indicating that cells with NF-κB activation and cells that were proliferating represented two largely independent populations (*Figure 5J*). Within the HSPC subsets, myeloid progenitor cells (L⁻S⁻K⁺) contained the largest fraction of rapidly proliferating cells, whereas HSCs and LSK cells contained the highest percentage of cells with NF-κB activation. This suggests that

NF-κB does not appear to directly stimulate HSPC proliferation in a cell-autonomous manner. However, it remains to be seen whether NF-κB regulates other aspects of HSC biology, including cell death, differentiation, and trafficking. Cell intrinsic function of NF-κB in HSCs requires further study.

## Proinflammatory cytokine IL-6 is a culprit NF-κB–responsive gene mediating HSC depletion and myeloproliferation

Because of the complexity and the overwhelmingly large list of NF-κB-responsive genes, we wanted to determine whether the HSC defect and the myeloproliferative pathology involve many NF-κB–activated genes acting together or if there are key culprit genes mediating the process.

Proinflammatory cytokines, such as IL-6 and TNFα, both of which are highly upregulated upon NF-κB activation, can be potent oncogenic factors, especially in epithelial cancers (*Naugler and Karin, 2008*). More importantly, overexpression of IL-6 in bone marrow cells in mice results in myeloproliferative or lymphoproliferative disease (*Brandt et al., 1990*; *Hawley et al., 1992*). Furthermore, we have shown that NF-κB appears to regulate HSC proliferation in miR-146a-deficient mice in a non-cell-autonomous manner. Therefore, we focused on the proinflammatory cytokines that were overproduced in miR-146a KO mice, believing that miR-146a-deficient mice likely suffer from a chronic inflammation-driven process, and these proinflammatory cytokines produced by other immune cells upon NF-κB activation may be the direct activators of HSCs.

IL-6 and TNFα are both upregulated in aging mR-146a KO mice (*Boldin et al., 2011*; *Zhao et al., 2011*). However, upregulation of TNFα was only prominent in miR-146a KO spleens that have developed myeloid sarcomas, but not in ones without overt tumors, suggesting that TNFα upregulation may be a quite late event in oncogenesis. To evaluate the temporal relationship between the increase in inflammatory cytokines and the onset of HSC exhaustion and myeloproliferation, we examined younger, 6- to 10-month-old, mice. At this time, the HSC depletion and myeloproliferative phenotype start to become prominent, but no overt splenic tumors have developed. Similar to what was observed in 18-month-old mice, IL-6 expression was upregulated in both spleen and bone marrow cells of miR-146a KO mice, compared to WT mice. More importantly, miR/p50 DKO mice showed a level of IL-6 comparable to that of WT mice (*Figure 6A,B*). However, the same trend was not observed for TNFα expression at this age (*Figure 6—figure supplement 1A*). Furthermore, when bone marrow-derived macrophages (BMMs) from 8-week-old mice were stimulated with LPS in vitro, IL-6, but not TNFα, showed consistently increased induction in miR-146a KO BMMs compared to WT BMMs. Interestingly, the exaggerated IL-6 induction in miR-146a KO BMMs was significantly more prominent with restimulation at a time when WT BMMs showed resistance to endotoxin restimulation (*Figure 6C*; *Nahid et al., 2011*). This again suggests that miR-146a may be particularly important during chronic and repeated inflammatory challenge. In addition, induction of IL-6, but not TNFα, in BMMs was highly dependent on p50. In p50 KO BMMs, IL-6 induction was almost completely abolished (*Figure 6C* and *Figure 6—figure supplement 1B*). These data suggest that IL-6 upregulation is an early feature in miR-146a deficiency driven HSC depletion and myeloproliferation and reduction in the IL-6 level may be an important factor underlying the reduced pathology when p50 is deleted.

To determine whether IL-6 is a culprit gene downstream of NF-κB-mediating HSC depletion and myeloproliferative disease, we bred miR-146a KO mice with mice knocked out for the *Il6* gene. *Mir146a⁻/⁻ Il6⁻/⁻* (miR/IL6 DKO) mice were born at the expected Mendelian frequency and appeared normal. Unperturbed young WT, miR-146a KO, IL6 KO, and miR/IL6 DKO mice showed similar levels of CD11b⁺ and Gr1⁺ myeloid cells in peripheral blood, while B cells were slightly reduced in miR/IL6 DKO mice (*Figure 6—figure supplement 1C*). When LPS was repeatedly administered in young WT, miR-146a KO, and miR/IL6 DKO mice, an accelerated myeloproliferative phenotype was observed in miR-146a KO mice that was largely absent in miR/IL6 DKO mice. Specifically, the significantly increased spleen weight, total white blood cells, myeloid cells, and erythroid precursor cells present in miR-146a KO were all reduced in miR/IL6 DKO mice to levels comparable to those of WT mice (*Figure 6D*). The same trend was also observed in the peripheral blood (*Figure 6—figure supplement 1D*). In the bone marrow, repeated LPS stimulation of WT and miR/IL6 DKO mice induced variable hypocellularity. However, the depletion was more severe and consistent in miR KO bone marrow (*Figure 6—figure supplement 1E*). Furthermore, when WT, miR-146a KO, IL6 KO, and miR/IL6 DKO mice were allowed to age to 6–7 months, myeloproliferation and marrow fibrosis observed in miR-146a KO mice were also significantly reduced in miR/IL6 DKO mice (*Figure 6E* and *Figure 6—figure supplement 1F,G*).

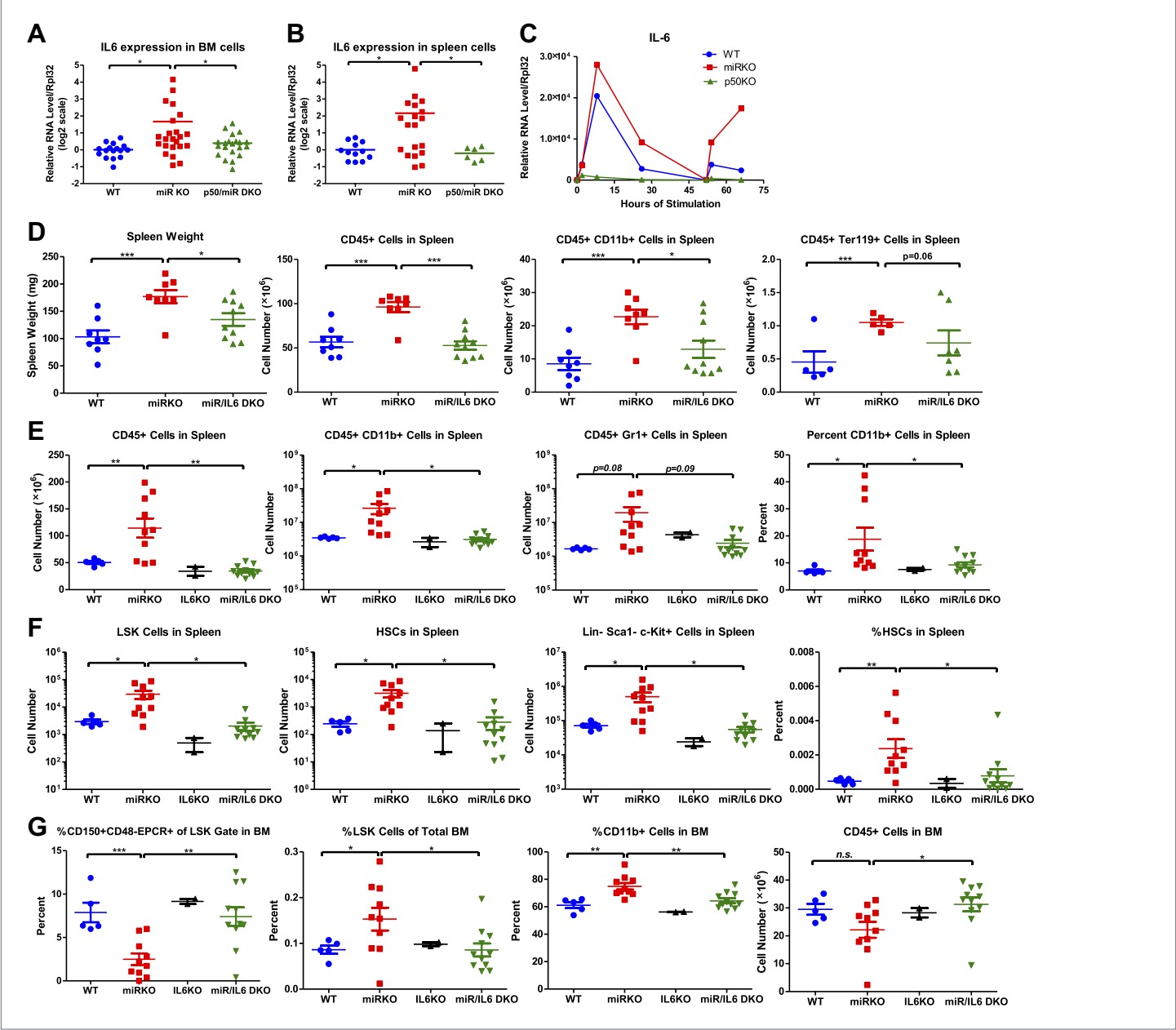

**Figure 6**. NF-κB-regulated pro-inflammatory cytokine IL-6 is an important driver of HSC depletion and myeloproliferation. Gene expression of IL-6 in bone marrow cells (BM) (**A**) and spleen cells (**B**) of aging WT, miR KO, and miR/p50 DKO mice measured by RT-qPCR. All mice are age- and sex-matched 6- to 10-month-old female mice. Gene expression of IL-6 (**C**) in bone marrow–derived macrophages (BMMs) stimulated in vitro with LPS (100 ng/ml) measured by RT-qPCR. First stimulation with LPS was given at 0 hr and restimulation at 48 hr. BMMs are generated from 8-week-old WT, miR KO, and p50 KO mice. (**D**) 2-month-old WT, miR KO, and miR/IL6 DKO mice after repeated intraperitoneal injection of LPS (3 mg LPS/kg body weight on day 1, 3, 5, and 7) and mice were harvested on day 8. Spleen weight and total number of CD45+, CD45+CD11b+, and CD45+ Ter119+ cells in spleen were shown. (**E**)–(**G**) Age- and sex-matched WT, miR KO, IL6 KO, and miR/IL6 DKO mice were allowed to age to 6–7 months before harvested for FACS analysis. (**E**) Quantification of number of CD45+, CD45+CD11b+, CD45+Gr1+, and percent of CD11b+ cells in spleen. (**F**) Quantification of number of HSPCs, including LSK cells, LSK CD150+CD48− HSCs and Lin−cKit+Sca1− myeloid progenitor cells, in spleen and percent of HSCs in spleen. (**G**) Quantification of percent of CD150+CD48−EPCR+ HSCs of LSK gate, LSK cells of total BM, and CD11b+ cells of total BM and total number of CD45+ cells in BM.

The following figure supplements are available for figure 6:

**Figure supplement 1**. NF-κB–regulated proinflammatory cytokine IL-6 is an important driver of HSC depletion and myeloproliferation.

More interestingly, the disrupted bone marrow HSPC homeostasis and expansion of splenic HSPCs seen in miR-146a KO mice were also partially normalized in miR/IL-6 DKO mice (*Figure 6F,G*). These data show that in the absence of IL-6, miR-146a KO mice display a partially reduced HSC defect and less myeloproliferative disease, indicating that upregulation of the NF-κB-responsive proinflammatory cytokine IL-6 is an important driver of the HSC defect and myeloproliferative disease in miR-146a KO mice under chronic inflammatory stress induced by aging or repeated bacterial stimulation.

## Cellular source and cellular target of IL-6

Because a variety of cells have the ability to produce IL-6 and IL-6 in turn has a pleiotropic effect on multiple cell types of the hematopoietic lineage, we wanted to identify the important cellular source and cellular target of IL-6 in mediating HSC defect and myeloproliferative disease in miR-146a KO mice. We have previously shown that miR-146a-deficient lymphocytes are involved in the development of the HSC defect and myeloproliferation (*Figure 4*), we asked whether overproduction of IL-6 by miR-146a-deficient lymphocytes represents one potential contributing mechanism. To test this, we first stimulated WT, miR-146a KO, Rag1 KO, and miR/Rag1 DKO mice with LPS and measured IL-6 production in vivo. We found increased IL-6 production in the serum of miR-146a KO mice, compared to that of WT mice. In comparison, the serum IL-6 level in miR/Rag1 DKO mice was only modestly increased in a nonstatistically significant manner, compared to that of Rag1 KO mice (*Figure 7A*). This indicates that in the absence of lymphocytes, exaggerated IL-6 production is attenuated. However, the modest increase in IL-6 production in miR/Rag1 DKO mice suggests that cells other than lymphocytes, such as myeloid cells shown earlier (*Figure 6C*), also exhibit enhanced IL-6 production in response to stimulation. To further study the contribution of lymphocytes, we stimulated splenocytes from WT or miR-146a KO mice in vitro with either LPS to activate both T and B cells or a combination of anti-CD3 and anti-CD28 to activate T cells specifically. In both conditions, we observed significantly increased production of IL-6 by miR-146a KO lymphocytes, compared to WT lymphocytes (*Figure 7B,C*). Lastly, to determine whether IL-6 overproduced by miR-146a KO T cells represents an important contributor to the HSC phenotype, we transplanted total T cells from spleens of WT, miR-146a KO, or miR/IL-6 DKO mice into 10-month-old miR/Rag1 DKO mice to see whether reintroducing miR-146a-deficient T cells will promote bone marrow depletion. Interestingly, compared to miR/Rag1 DKO mice that received WT T cells, mice that received miR-146a KO T cells, but not miR/IL-6 DKO T cells, displayed bone marrow depletion of total white blood cells, myeloid progenitor cells, LSK cells, and long-term HSCs (*Figure 7D*). In addition, mild leukopenia was also observed only in the mice that received miR-146a KO T cells (*Figure 7E*). These data have demonstrated the particular importance of miR-146a-deficient T cells, specifically their dysregulated IL-6 production, in driving HSC and bone marrow depletion. However, because miR/Rag1 DKO mice do exhibit depleted HSCs (*Figure 4*), other intrinsic and extrinsic contributors of HSC defect besides T cells require further examination. In addition, it remains to be tested whether the compositions and inflammatory properties of T cells, such as percentages of regulatory T cells, $T_h1$, $T_h2$, and $T_h17$ cells, from WT, miR-146a KO, and miR/IL-6 DKO mice are different, which may explain their different effects on promoting inflammation and bone marrow depletion.

To understand whether IL-6 impacts hematopoiesis through a direct effect on HSPCs, we first stimulated a 1:1 mixture of WT or miR-146a KO CD45.2$^+$ cKit$^+$ bone marrow cells and WT CD45.1$^+$ cKit$^+$ bone marrow cells with IL-6 or LPS for 3 days. We observed increased percentages of both cKit$^+$ and CD11b$^+$ KO cells in the coculture, showing that miR-146a KO cKit$^+$ cells have a proliferative/survival advantage and enhanced myeloid differentiation over WT cKit$^+$ cells under IL-6 or LPS stimulation (*Figure 7F*). To further determine the direct effect of IL-6 and LPS on more refined HSPC subsets, we sorted LSK cells and long-term HSCs from bone marrow of young WT and miR-146a KO mice and stimulated them with IL-6 or LPS in the presence of BrdU for 18 hr. While HSCs did not have statistically significant difference in BrdU incorporation, KO LSK cells showed higher percentage of BrdU$^+$ cells than WT LSK cells under IL-6 stimulation (*Figure 7G,H*). Interestingly, no proliferative differences between WT and KO LKS cells or HSCs were detected in response to a single dose of LPS stimulation at 1 μg/ml concentration (*Figure 7G,I*). These data suggest that the hematopoietic phenotypes seen in miR-146a KO mice may be mediated by IL-6 acting directly on LSK cells, but not long-term HSCs. Furthermore, the in vivo effect of LPS on miR-146a KO mice in inducing enhanced myeloproliferation and HSPC proliferation is at least partly through stimulating the production of cytokines such as IL-6.

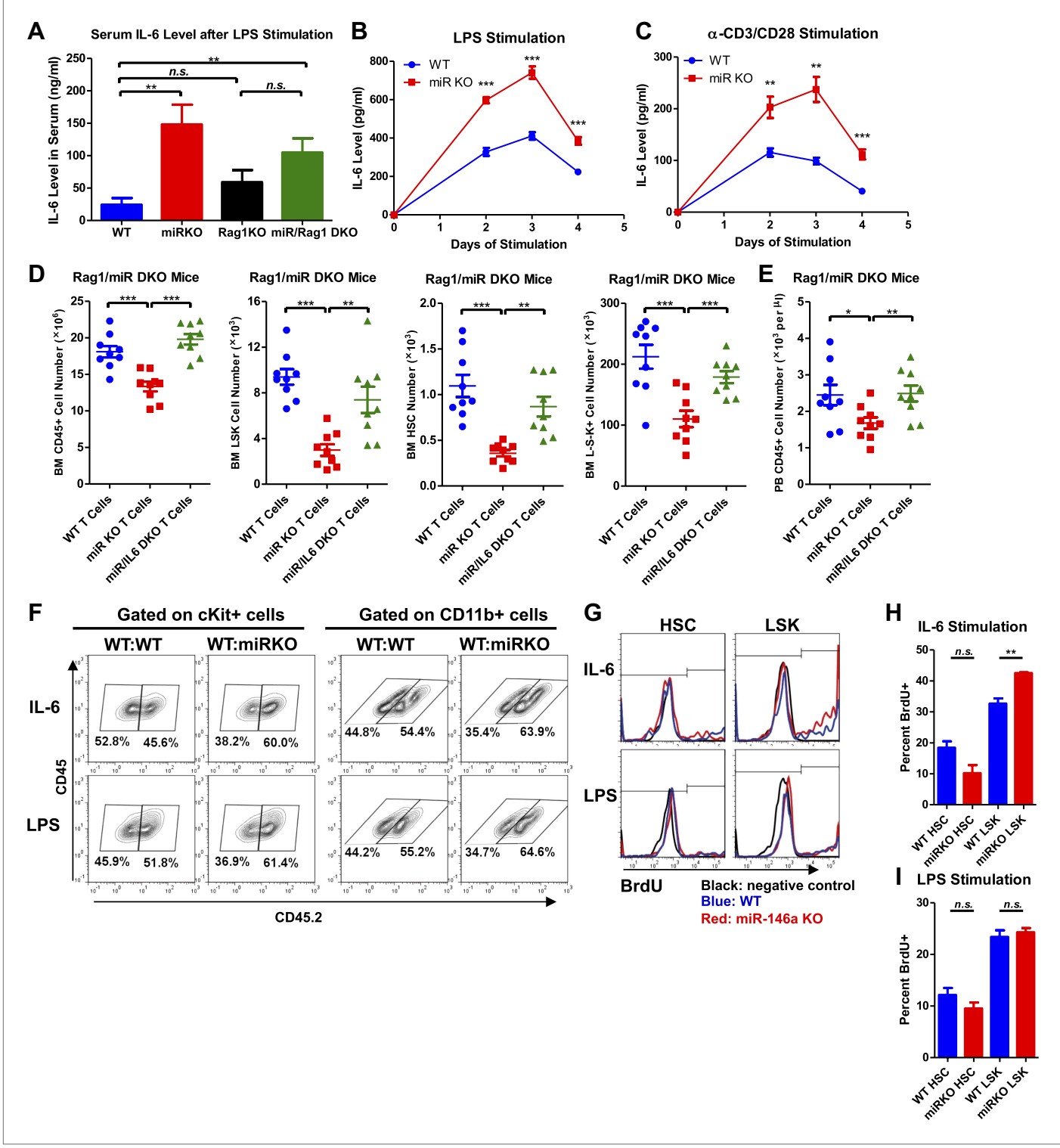

**Figure 7**. Analysis of cellular source and direct cellular target of IL-6. Serum level of IL-6 measured by ELISA in 2-month-old WT, miR KO, Rag1 KO, and miR/Rag1 DKO mice stimulated with LPS (1 mg LPS /kg body weight) intraperitoneally for 6 hr (**A**) IL-6 concentration measured by ELISA in the culture medium of splenocytes stimulated in vitro with LPS (10 μg/ml) (**B**) or anti-CD3 (1 μg/ml)/anti-CD28 (0.5 μg/ml) antibodies (**C**) for 4 days. (**D**) and (**E**) CD3ε⁺ T cells were purified from spleens of 10-month-old WT, miR-146a KO (miR KO), or miR/IL-6 DKO mice. 4 million T cells per mouse were transplanted into 10-month-old miR/Rag1 DKO mice intravenously. miR/Rag1 DKO mice were harvested 1 month after transplant for FACS analysis of white blood cells and HSPCs of bone marrow (**D**) and/or peripheral blood (**E**). (**F**) cKit⁺ cells were purified from bone marrow of 8-week-old CD45.1⁺ WT and CD45.2⁺ WT

*Figure 7. Continued on next page*

*Figure 7. Continued*

or miR-146a KO mice. A 1:1 mixture of CD45.1 WT/CD45.2 WT or CD45.1 WT/CD45.2 KO cKit$^+$ cells were co-cultured under IL-6 (50 ng/ml) or LPS (100 ng/ml) stimulation for 3 days. Percentages of CD45.2$^+$ cKit$^+$ or CD11b$^+$ were analyzed by FACS. (**G**)–(**I**) LSK cells or long-term HSCs (LSK CD150$^+$CD48$^-$) were sorted from 8-week-old WT or miR-146a KO mice and were cultured in separate wells with IL-6 (50 ng/ml) or LPS (1 μg/ml) stimulation in the presence of BrdU (50 μM). After 18 hr, cells were analyzed for cell surface marker expression and BrdU incorporation by FACS. Representative FACS histograms of BrdU status of HSCs or LSK cells. Negative control represents identically gated and stained cells in the absence of BrdU pulse (**G**). Quantification of percent BrdU$^+$ HSCs or LSK cells under IL-6 (**H**) or LPS (**I**) stimulation.

## Upregulation of the miR-146a target, TRAF6, results in bone marrow failure

Two of the best-validated miR-146a targets, TRAF6 and IRAK1, are signal transduction proteins upstream of NF-κB activation. To determine whether increased expression of TRAF6 and/or IRAK1 is responsible for the observed HSC exhaustion in the absence of miR-146a, we first measured whether their expression was derepressed in miR-146a-deficient bone marrow cells. BMMs from miR-146a KO and WT mice were stimulated with LPS for 48 hr and then restimulated with a second dose for an additional 16 hr. The transcript level of TRAF6 showed consistent derepression in miR-146a KO BMMs throughout stimulation, whereas the transcript level of IRAK1 showed perhaps an oscillating pattern but was not consistently higher than that of WT BMMs (*Figure 8A,B*). Furthermore, when HSPCs from young WT and miR-146a KO mice were analyzed for gene expression, derepression of TRAF6 and IRAK1 were quite modest in bulk bone marrow cells but were highest in long-term HSCs, suggesting that miR-146a may play a particularly important repressive role in HSCs, acting on TRAF6 and IRAK1 (*Figure 8C*).

To determine the functional consequence of upregulating TRAF6 and IRAK1, we used a GFP-expressing retroviral vector to overexpress TRAF6 (pMIG-TRAF6) or IRAK1 (pMIG-IRAK1) in bone marrow cells enriched for HSPCs by 5-fluorouracil (5-FU) treatment. A vector expressing an irrelevant protein, luciferase (pMIG-Luc), was used as the control. The vectors can consistently overexpress TRAF6 and IRAK1 by 10-fold and 100-fold, respectively, in bone marrow HSPCs (with about 50% transduction efficiency) (*Figure 8—figure supplement 1A*). After transplantation, we followed the mice for 9 months. Interestingly, the percent of GFP$^+$ cells in nucleated peripheral blood cells transduced with either pMIG-Luc or pMIG-IRAK1 remained stable or increased slightly, whereas the GFP percentage in the pMIG-TRAF6 group declined from over 40% initially to less than 10% by the end of 9 months (*Figure 8D*). Histological analysis of harvested femur bones revealed that TRAF6-expressing mice had reduced bone marrow cellularity, compared to the control mice (*Figure 8E*). Given the lack of apparent phenotype in IRAK1-overexpressing mice, we further pursued TRAF6 as the more relevant target of miR-146a in the context of HSC biology. When we repeated the bone marrow transplant with 100% transduced bone marrow cells by sorting the GFP$^+$ cells, mice receiving TRAF6-overexpressing bone marrow cells rapidly succumbed to bone marrow failure (*Figure 8F*). These mice showed severe anemia and reduced numbers of GFP$^+$ white blood cells in their peripheral blood, compared to the pMIG-Luc control group (*Figure 8G–I*). These data indicate that upregulation of TRAF6 in HSPCs results in bone marrow failure in mice, thus emphasizing the importance of tightly regulating TRAF6 level in HSPCs. This result suggests that derepression of TRAF6 in miR-146a-deficient HSPCs could be primarily responsible for driving HSC depletion and bone marrow failure.

## Downregulation of miR-146a in human MDS samples

Myelodysplastic syndromes (MDS) represent a group of human hematopoietic malignancies that are thought to originate from HSCs (*Nimer, 2008a*). HSCs in MDS patients have a defect in the ability to differentiate into mature cells, leading to peripheral cytopenia. MDS also have a predilection to progress to bone marrow failure or acute myelogenous leukemia (AML). MiR-146a KO mice recapitulate several key characteristics of MDS, including the decline in function of HSCs, peripheral cytopenia, and the propensity to progress to bone marrow failure and myeloid malignancy. To assess whether miR-146a deficiency might represent a pathogenic feature of MDS, we analyzed the expression of miR-146a in bone marrow samples from healthy donors, MDS, and AML patients. The cohort of unselected MDS, but not AML, samples showed a fourfold reduction in the level of miR-146a (*Figure 8J*), suggesting that miR-146a deficiency may be involved in MDS pathogenesis (*Starczynowski et al., 2010*; *Rhyasen and Starczynowski, 2012*).

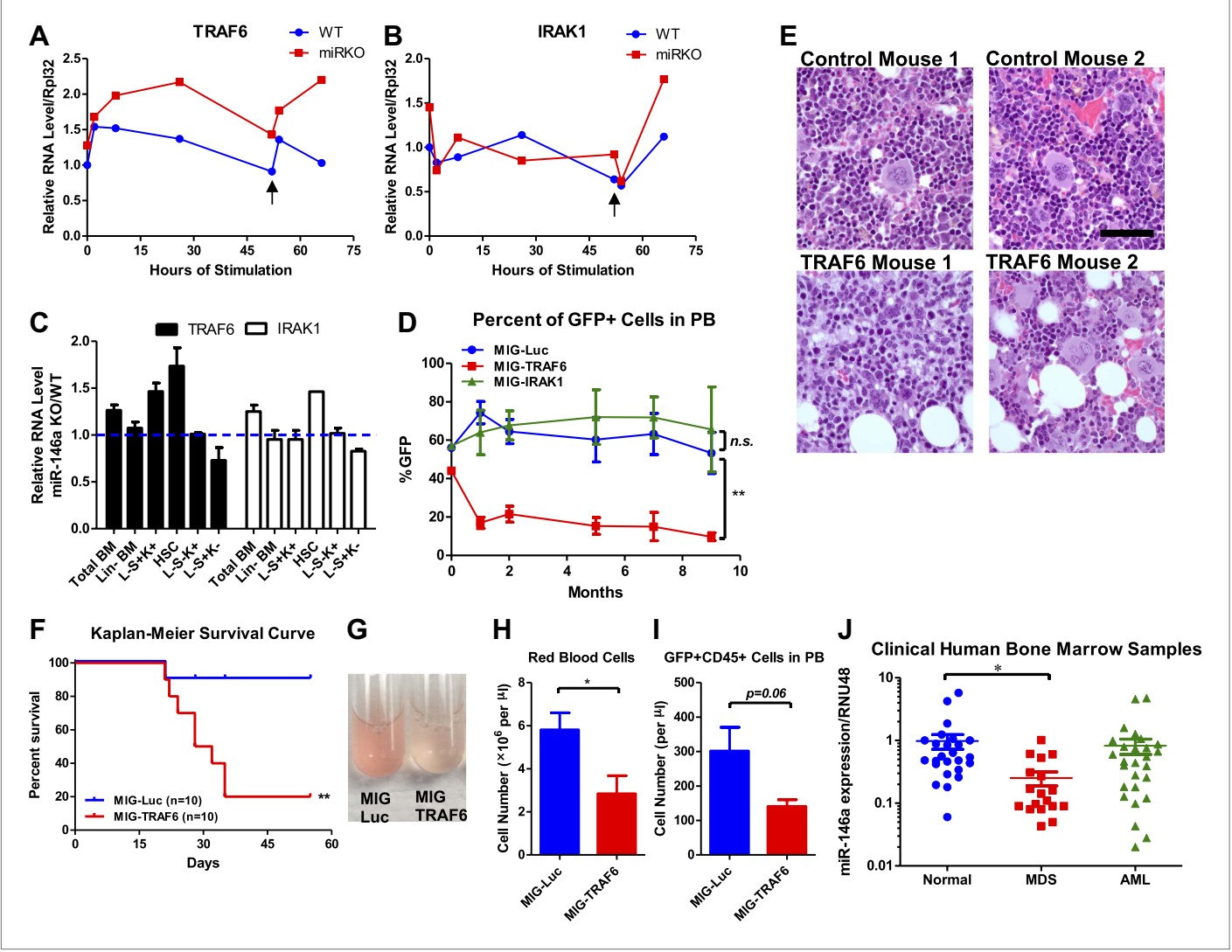

**Figure 8**. Derepression of TRAF6, a miR-146a target, is responsible for bone marrow failure. Transcript levels of TRAF6 (**A**) and IRAK1 (**B**) in WT and miR-146a KO (miR KO) bone marrow–derived macrophages (BMMs) stimulated with LPS, which was added to the culture medium at 0 and 48 hr (black arrow). (**C**) Transcript levels of TRAF6 and IRAK1 in total BM, Lin⁻ BM, and FACS-sorted LSK cells, LSK CD150⁺CD48⁻ HSCs, L⁻K⁺S⁻ cells, and L⁻K⁻S⁺ cells from 8-week-old WT and miR-146a KO mice. Fold change of miR-146a KO over WT cells was graphed. (**D**) and (**E**) BM HSPCs overexpressing luciferase (MIG-Luc), TRAF6 (MIG-TRAF6), or IRAK1 (MIG-IRAK1) were transplanted into lethally irradiated WT recipient mice. Transduction efficiency was about 50% in all groups as measured by FACS before transplantation. (**D**) Percent of GFP⁺ cells in transduced HSPCs before transplantation and in peripheral blood of reconstituted mice at month 1, 2, 5, 7, and 9 were analyzed by FACS. (**E**) Representative photographs of histological analysis (H&E stain) of femur bones of MIG-Luc control and MIG-TRAF6 mice harvested 9-month after transplantation. Scale bar, 40 µm. (**F**)–(**I**) BM HSPCs overexpressing luciferase (MIG-Luc) or TRAF6 (MIG-TRAF6) were transplanted into lethally irradiated WT recipient mice. Transduced HSPCs were sorted for GFP expression to ensure the transplanted HSPCs were 100% GFP⁺. (**F**) Kaplan–Meier survival curve of WT recipient mice reconstituted with BM HSPCs overexpressing luciferase (MIG-Luc) or TRAF6 (MIG-TRAF6). Peripheral blood (PB) analysis of MIG-Luc and MIG-TRAF6 mice at 1 month after transplantation. (**G**) Representative photograph of 1:1000 diluted PB in phosphate-buffered saline (PBS). Red blood cells in PB were counted with hemocytometer (**H**) and total number of GFP⁺CD45⁺ cells in PB (**I**) were measured by FACS. (**J**) Downregulation of miR-146a in human myelodysplastic syndromes (MDS) samples. Expression level of miR-146a in bone marrow samples from healthy donors (normal), MDS and acute myelogenous leukemia (AML) patients by Taqman RT-qPCR. RNU48 was used as the normalization gene.

The following figure supplements are available for figure 8:

**Figure supplement 1**. Gene expression analysis of miR-146a targets in BM HSPCs.

## Discussion

In this study, we have shown that a single microRNA, miR-146a, is a critical regulator of HSC homeostasis following chronic exposure of mice to LPS or during the aging process. Mice lacking miR-146a develop a series of defects that can be accelerated by multiple LPS treatments. Extrapolating from this observation, we suggest that the natural exposure of miR-146a KO mice to bacteria and other infectious agents may be the cause of the 'spontaneous' deterioration of HSC function (bone marrow failure), excessive myelopoiesis, and tumor formation that are observed in these animals. It could well be that the excess myelopoiesis seen in normal mice as they age is a consequence of the same process (*Salminen et al., 2008*; *Beerman et al., 2010*). One obvious test of these notions, to examine HSC function in germ-free mice, is not ideal because of the many secondary consequences of completely ablating the normal microbiota and will require more subtle approaches. The remarkable circumstance that a single microRNA plays such a crucial role in this process has allowed us to uncover it.

Because of the existence of mice lacking miR-146a, we were able to use mouse genetics to elucidate the pathway by which miR-146a functions. We propose the following pathway of action of miR-146a: infections or other perturbations activate NF-κB through TRAF6 and other signal transducers; the basal and NF-κB-enhanced levels of miR-146a then limit the amount of TRAF6, allowing the system to quickly return to basal NF-κB activity when the danger has passed; the transient NF-κB activation causes a transient increase in IL-6 and the IL-6 acts on the stem and progenitor cells to transiently promote proliferation and myeloid differentiation. Lacking miR-146a, the action of NF-κB is extended, the production of IL-6 is exaggerated, the myelopoietic stress on the HSPCs is extended, and the over time the pathological symptoms of continual myelopoiesis, bone marrow failure, and cancer emerge. This may be a pathway in humans that leads to myelodysplastic syndrome.

We have demonstrated the importance of the pathway consisting of miR-146a/TRAF6/NF-κB/IL-6 in regulating HSCs during chronic inflammation at the organismal level. However, because HSCs are bathed in a complex cytokine environment and engaged in a complex interaction with other hematopoietic and nonhematopoietic cells, there are a number gaps in our knowledge that requires further study to fully disentangle which molecules are functioning cell autonomously in HSCs and which ones are working through other hematopoietic cells to indirectly affect HSCs. In addition, we have to be cautious of claims about HSC-intrinsic function due to the fact that HSCs are continuously producing more differentiated cells, which harbor the same genetic alterations and are also subject to the same external stimuli.

In our effort to understand the function of each molecule in a cell-specific manner, we have undertaken various in vivo transplant experiments and in vitro cell culture studies. We have shown that miR-146a has an intrinsic function within HSCs because miR-146a-deficient HSCs have an intrinsic defect and are preferentially depleted compared to WT HSCs in the same environment. However, this intrinsic defect is modest. Other miR-146a-deficient hematopoietic cells, especially T cells, have a strong effect on HSCs. In addition, miR-146a-deficient nonhematopoietic cells also contribute to the overall HSC abnormality. TRAF6 is derepressed in miR-146a-deficient HSPCs and overexpression of TRAF6 in enriched HSPCs recapitulates the bone marrow failure phenotype. However, whether TRAF6 is functioning in HSCs or more differentiated cells require further examination. Furthermore, TRAF6 is known to regulate the MAPK and PI3K pathways in addition to NF-κB. Further work will be required to clarify their respective contributions in mediating bone marrow failure. We have also demonstrated the functional activity of NF-κB in HSCs and other hematopoietic cells. However, NF-κB does not appear to regulate HSC proliferation cell autonomously. In fact, in WT mice, NF-κB activation does not correlate with proliferative activity in any stem and progenitor cells. The functional consequence of NF-κB activation within HSPCs remains an interesting unanswered question. What is clear is that NF-κB does a significant amount of its 'damage' to HSCs through the proinflammatory cytokine IL-6, produced by T cells and myeloid cells, among others. IL-6 acts directly on LSK cells to stimulate proliferation and miR-146a–deficient LSK cells are more proliferative than WT LSK cells under IL-6 stimulation. In summary, miR-146a is functional in HSCs, LSK cells, T cells, myeloid cells, as well as nonhematopoietic cells, and deficiency of miR-146a in all these cells leads to the full-blown myeloproliferative disease and HSC exhaustion during chronic inflammatory stress. Based on this study and our previous work (*Boldin et al., 2011*; *Zhao et al., 2011*; *Yang et al., 2012*), we have shown that a

circuitry involving miR-146a/TRAF6/NF-κB is operational in miR-146a-deficient T cells and myeloid cells, leading to IL-6 overproduction. IL-6 in turn directly stimulates LSK cells to proliferate and differentiate. We have also demonstrated derepression of TRAF6 and enhanced activation of NF-κB in miR-146a-deficient HSCs and LSK cells. However, it is not yet clear that the cell-autonomous function of miR-146a in HSCs is through the regulation of TRAF6 and NF-κB and the functional importance of TRAF6 and NF-κB in HSCs requires further clarification.

The function of individual miRNAs in the most primitive hematopoietic stem cells is relatively unexplored, except in the case of miR-125 family, which is enriched in the long-term HSCs and can positively regulate HSC number and long-term HSC output (*Guo et al., 2010*; *O'Connell et al., 2010*). In this study, we added miR-146a to the list of genes that regulate HSC homeostasis during chronic inflammation and aging. Overexpression of miR-155 also causes pathological myelopoiesis similarly to that caused by deletion of miR-146a. MiR-155 appears to act by inhibiting the negative-acting phosphatase SHIP1 (*O'Connell et al., 2009*). Enforced expression of TRAF6 in HSPCs results in cell death and/or engraftment failure and rapid bone marrow failure, a finding consistent with a previous study (*Starczynowski et al., 2010*). The same study also shows that some key features of 5q-syndrome, a subtype of MDS, seen in TRAF6-overexpressing mice are mediated by IL-6. It is worth noting that the level of TRAF6 expression in transduced HSPCs is about 10-fold higher than that observed in miR-146a-deficient HSCs, and thus the phenotype of enforced TRAF6 overexpression in HSPCs is more rapid and dramatic, precluding a careful analysis of HSC function and activity. It will be interesting to develop a system to modulate the level of TRAF6 overexpression in a more controlled manner for HSC analysis and to characterize the effects of downregulating TRAF6 genetically or by RNA interference in miR-146a-deficient HSCs. In addition to TRAF6 and IRAK1, STAT1 and RelB have been identified as direct targets of miR-146a in regulatory T cells and Ly6C$^{hi}$ monocytes, respectively (*Lu et al., 2010*; *Etzrodt et al., 2012*). Recent studies have provided strong evidence for roles of both IFNα and IFNγ in regulating HSC quiescence and proliferation (*Essers et al., 2009*; *Baldridge et al., 2010*). As a downstream transcription factor of both types of interferons, one might speculate that derepression of STAT1 in miR-146a-deficient HSCs may have a cell-intrinsic role in promoting HSC proliferation and exhaustion. Our preliminary analysis has shown upregulation of STAT1 mRNA in miR-146a-deficient HSPCs (*Figure 8—figure supplement 1B*). A recent study has also suggested a role of noncanonical NF-κB subunits RelB and NF-κB2 in regulating HSC self-renewal (*Zhao et al., 2012*). It remains to be determined whether other putative targets of miR-146a, such as STAT1 and RelB, also contributes to the HSC defect in the context of miR-146a deficiency.

In an effort to characterize the effects downstream of NF-κB activation in the absence of miR-146a, we identified IL-6 as one of the culprit NF-κB-responsive genes. Among the pleiotropic functions, IL-6 and its downstream transcription factor STAT3 are important regulators of emergency granulopoiesis (*Zhang et al., 1998*, *Zhang et al., 2010*). In addition, IL-6 can also activate NF-κB in certain contexts (*Iliopoulos et al., 2009*). So, IL-6 can exert its effect either directly on HSPCs and myeloid cancer cells or indirectly by contributing to the general proinflammatory and proproliferative environment through both STAT3 and NF-κB. This signaling pathway involving NF-κB/IL-6/STAT3 has been demonstrated to be important in the pathogenesis of various epithelial cancers associated with inflammation, especially gastrointestinal cancer and breast cancer (*Naugler and Karin, 2008*; *Grivennikov et al., 2009*; *Iliopoulos et al., 2009*; *Rakoff-Nahoum and Medzhitov, 2009*). Interestingly, a recent report shows that chronic myelogenous leukemia (CML) driven by the classic Bcr-Abl oncogenic fusion protein can be attenuated by genetic deletion of *Il6* (*Reynaud et al., 2011*). It will be interesting to identify the genes regulated by NF-κB and STATs that are directly responsible for activating the proliferation and differentiation programs within HSPCs.

This study has also identified an important role of dysregulated lymphocytes in driving HSC abnormality and myeloproliferative disease in our model, suggesting that autoreactive lymphocytes can provide extrinsic stimulus to diseases like bone marrow failure and myeloproliferative neoplasms/myelodysplastic syndromes in genetically susceptible hosts. Importantly, systemic autoimmunity has long been observed in some MDS patients (*Nimer, 2008b*). In addition, given the importance of IL-6, a known regulator of CD4 T cell differentiation toward either $T_h17$ or $T_{reg}$ cells, it will be interesting to determine whether overproduction of IL-6 contributes to abnormal hematopoiesis by altering the ratio of CD4 T cell subsets. Defining the precise role of various lymphocyte subsets, including $T_h1$, $T_h2$, $T_h17$, and $T_{reg}$ cells, using lineage-specific miR-146a conditional knockout mouse

may provide further insight into how different immune cells influence physiological and pathological hematopoiesis.

This study provides an insight into the function of miR-146a in regulating HSC proliferation and differentiation under the influence of physiological stressors. Given the multitude of mechanisms involved in downregulating NF-κB activity (*Liew et al., 2005*; *Ruland, 2011*), these results highlight the critical, nonredundant role of miR-146a in the negative regulation of NF-κB activity during chronic inflammation. In this way, miR-146a acts as a guardian of the functional capabilities and longevity of murine hematopoietic stem cells. In addition, this study also has several other important implications. First, it provides direct evidence that prolonged and uncontrolled inflammation-driven hematopoiesis can ultimately exhaust the HSC pool and lead to myeloid malignancies in genetically susceptible hosts, according nicely with some recent large scale epidemiological studies showing that chronic immune stimulation from past infection or autoimmunity increases the risk of developing myeloid malignancies, including AML, MDS, and myeloproliferative neoplasms (MPN) (*Anderson et al., 2009*; *Kristinsson et al., 2011*; *Hasselbalch, 2012*). This study provides a possible molecular basis for these intriguing epidemiological observations and offers a unique experimental system to further explore the cellular and molecular pathways by which infection and autoimmunity can trigger myeloid malignancies; it also provides a system to test potential therapeutic interventions and chemoprevention. Second, miR-146a-deficient mice represent an excellent model to understand the pathogenesis of MDS, a hematopoietic malignancy of older adults (median age of 70 years) (*Sekeres et al., 2008*) that has consistently shown reduced expression of miR-146a. Lastly, it suggests that chronic inflammation may be a potential cause of the age-related decline in HSC function. Therapeutically, given that many of the cellular and molecular components are important in driving the overall pathology, inhibition of p50 subunit of NF-κB, hyperactivated lymphocytes, IL-6 overproduction, and TRAF6 represent multiple opportunities for therapeutic intervention to disrupt the pathogenic process leading to myeloid malignancies, and combinatorial inhibition may have even greater therapeutic impact.

## Materials and methods

### Mice

All mice were on a C57BL/6 genetic background and housed under specific pathogen-free condition at the California Institute of Technology. All double knockout mice were made by crossing single knockout mice. Experiments with mice were approved by the Institutional Animal Care and Use Committee of the California Institute of Technology. Mouse harvest, tumor analysis, and tumor transplant into $Rag2^{-/-}$ $Il2rg^{-/-}$ mice were performed as described (*Zhao et al., 2011*). For in vivo and in vitro stimulation, *Escherichia coli* 055:B5 LPS (Sigma, St. Louis, MO) was used.

### Cell culture and bone marrow–derived macrophages (BMMs)

Total bone marrow cells from wild-type, miR-146a$^{-/-}$, or p50$^{-/-}$ mice were lysed with red blood cell lysis buffer (Biolegend, San Diego, CA) and were cultured in DMEM supplemented with 10% (vol/vol) Fetal Bovine Serum (Cellgro, Manassas, VA), penicillin and streptomycin, and M-CSF (20 ng/ml) for 6 days. On day 7, BMMs were stimulated with *E. coli* 055:B5 LPS (100 ng/ml) for 0, 2, 8, and 24 hr. At 24 hr, BMMs were washed with phosphate-buffered saline (PBS) and taken off LPS stimulation for 24 hr. At 48 hr, BMMs were restimulated with 100 ng/ml LPS for additional 16 hr.

### Flow cytometry and sorting

Spleen, bone marrow, and peripheral blood cells were lysed with red blood cell lysis buffer. Fluorophore- or biotin-conjugated antibodies against CD45, CD3ε, CD4, CD8, CD11b, Gr1, B220, CD19, Ter119, NK1.1, cKit, Sca1, CD48, CD150, EPCR, and Ki-67 (Biolegend, San Diego, CA or eBioscience, San Diego, CA) were used for staining. BrdU staining was performed with BrdU staining kit from BD Biosciences. Cells were analyzed on a MACSQuant9 or MACSQuant10 Analyzer (Miltenyi, Auburn, CA) for both percentage and cell number. Data analysis was performed with FloJo software (TreeStar, Ashland, OR). Hematopoietic stem and progenitor cell sorting was performed by first depleting lineage$^+$ bone marrow cells with magnetic beads (Miltenyi) and then stained with indicated antibodies before sorted on a FACSAria machine (BD, Franklin Lakes, NJ). Analysis of BrdU incorporation and Ki-67 expression on purified LSK cells and HSCs in vitro was done with carrier cells to minimize cell loss during staining and permeabilization as previously described (*Mayle et al., 2013*).

## Quantitative RT-PCR and ELISA

Total RNA was extracted with TRIzol reagent (Invitrogen, Carlsbad, CA) from spleen or bone marrow cells after red blood cell lysis. cDNA was synthesized using iScript cDNA synthesis kit (Bio-Rad) followed by SYBR Green-based quantatitive PCR (Quanta Biosciences, Gaithersburg, MD). Rpl32 was used as the normalization gene. MiRNA detection was performed with Taqman RT-qPCR probes (Life Technologies, Carlsbad, CA). Sno202 was used as the normalization gene. Specific cytokines, including IL-6 and TNFα, were measured in cell culture medium or mouse serum by ELISA according to manufacturer's protocol (eBioscience).

## Histopathology

Organs were fixed in 10% neutral-buffered formalin immediately after necropsy. After fixation, organs was embedded in paraffin and processed for hematoxylin and eosin (H&E) staining. The histopathological analysis was performed by a board-certified hematopathologist.

## Statistical analysis

All figures were graphed as mean ± standard error of the mean (SEM). Student t-test and Kaplan–Meier survival analysis were performed using GraphPad Prism software. In all figures, * denotes $p<0.05$, ** denotes $p<0.01$, *** denotes $p<0.001$.

## Acknowledgements

We thank the staff at the California Institute of Technology animal facility and FACS Core sorting facility for excellent technical support and Drs Alejandro Balazs and Devdoot Majumdar for helpful discussion. NFκB-GFP reporter mice were generously provided by Dr Christian Jobin of the University of North Carolina, Chapel Hill.

## Additional information

### Competing interests

DB is a member of the board of directors of Regulus Therapeutics Inc., a company developing microRNA-based therapeutics. The other authors declare that no competing interests exist.

### Funding

| Funder | Grant reference number | Author |
| --- | --- | --- |
| National Institute of Allergy and Infectious Diseases | R01AI079243 | Jimmy L Zhao, Yvette Garcia-Flores, David Baltimore |
| National Heart, Lung, and Blood Institute | F30HL110691 | Jimmy L Zhao |
| National Cancer Institute | K08CA133521 | Dinesh S Rao |
| National Heart, Lung, and Blood Institute | R00HL102228 | Ryan M O'Connell |

The funders had no role in study design, data collection and interpretation, or the decision to submit the work for publication.

### Author contributions

JLZ, Conception and design, Acquisition of data, Analysis and interpretation of data, Drafting or revising the article, Contributed unpublished essential data or reagents; DSR, RMO, Conception and design, Acquisition of data, Analysis and interpretation of data; YG-F, Conception and design, Acquisition of data; DB, Conception and design, Analysis and interpretation of data, Drafting or revising the article

### Ethics

Animal experimentation: This study was performed in strict accordance with the recommendations in the Guide for the Care and Use of Laboratory Animals of the National Institutes of Health. All of the animal experiments were approved by the institutional animal care and use committee (IACUC) of the California Institute of Technology under protocol #1558.

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
