## [Decision Letter]

Thank you for choosing to send your work entitled “MicroRNA-146a acts as a guardian of the quality and longevity of hematopoietic stem cells” for consideration at *eLife*. Your article has been favorably evaluated by a Senior editor and 3 reviewers, one of whom is a member of our Board of Reviewing Editors.

The Reviewing editor and the other reviewers discussed their comments before we reached this decision, and the Reviewing editor has assembled the following comments to help you prepare a revised submission.

Zhao et al. characterize the effects of microRNA-146a (miR-146a) deletion on hematopoietic stem cell (HSC) function. Previous papers from this lab have shown that mir-146a deletion results in T-cell mediated chronic inflammation and a myeloproliferative disorder. In this work, the authors show that HSCs become depleted by 8 months after birth. HSCs have reduced function in transplantation assays, and they cycle more than wild-type HSCs. These defects are to a large extent mediated by lymphocytes and by activated NF-kB signaling. They also suggest a role for IL-6 in mediating HSC depletion downstream of NF-kB and that mir-146a regulates NF-kB activity through TRAF6. Their data are consistent with, but more mechanistic than, several other papers in the literature suggesting that inflammation drives HSCs into cycle and impairs HSC function. The miR-146a HSC phenotype is very interesting, supported by considerable high quality data, and arguably different from anything seen previously.

1) The authors summarize a model in the Discussion in which miR-146a acts cell-autonomously to activate NF-kB in HSCs by decreasing TRAF6 expression. This leads to IL-6 expression, which then promotes HSC depletion through increased myeloid differentiation. This is highly speculative and blurs the distinction between what Zhao et al. have tested and what they have not. Is the modest increase in TRAF6 really mediating the effects of miR-146a on HSCs? Does TRAF6 act autonomously in HSCs? What cells secrete IL-6? Does IL-6 really act on HSCs, or on other cells? Can IL-6 really promote myeloid differentiation by HSCs, or does it just promote the proliferation of myeloid progenitors that arise from HSCs? The authors have provided a lot of interesting and compelling data and should not be expected to answer all of these questions in this study; however, they should be careful to distinguish between what they have shown and what is speculation in the Discussion.

2) Is it really clear that mir-146a is acting cell-autonomously in HSCs? When small numbers of HSCs were transplanted with wild-type marrow, they did not observe HSC depletion in the recipient mice (Figure 2—figure supplement 1), and in competitive whole bone marrow transplants HSC depletion was not observed until 10 months after the transplant (Figure 2). Furthermore, they show that Rag deficiency partially rescues HSC depletion (Figure 4), raising the possibility that engrafted inflammatory T-cells or neoplastic myeloid cells indirectly cause HSC depletion. If only the mir-146a deficient HSCs are depleted in the competitive reconstitution experiments (not the competing wild-type HSCs) then this possibility would seem far-fetched. However, the data are usually presented in unconventional ways for the reconstitution experiments, making it hard to assess whether the competing wild-type HSCs consistently fail to be depleted. It would be very helpful to know absolute numbers of mir-146a deficient versus wild-type HSCs in the same mice.

3) The authors could more directly test whether NF-kB mediates HSC depletion cell-autonomously by transplanting wild-type and p50-deficient sorted HSCs with mir-146a-deficient competitor marrow. If the donor HSCs become depleted, the effects are likely non-cell autonomous to HSCs and they are an indirect consequence of other mir-146a deficient hematopoietic cells. If p50-deficient HSCs are not depleted even as the mir-146a competitor marrow causes inflammation and MPD, then NF-kb likely causes depletion cell-autonomously as the authors have concluded.

4) It is possible that the bone marrow HSCs in mir-146a deficient mice are not being depleted at all but are just being mobilized to the spleen. The authors need to look at the total number of HSCs in the bone marrow and the spleen to determine whether there is actually a decline in the total number of HSCs.

5) Even if there is a decline in the total number of HSCs, it is not clear this reflects “accelerated differentiation” into myeloid cells. The authors cite the increase in LSK cells as evidence for accelerated differentiation. However, there are many alternative possible explanations that have not been addressed. For example, LSK cells might proliferate at a higher rate (Figure 4) or the differentiation or phenotype of cells other than HSCs may be altered. Given that the idea of accelerated HSC differentiation is speculation, the authors should not present it as a tested conclusion.

6) In Figure 5, the authors show (using the NF-kB reporter strain) that ∼2x more miR-146a KO cells have active NF-kB signaling and argue that NF-kB regulates HSC proliferation. Although miR-146a/NF-kB double ko mice have nearly normal numbers of HSCs, it is still indirect evidence that NF-kB regulates HSC proliferation. In Figure 5 the authors show that miR-146a ko cells proliferate more strongly. It would be of interest for the authors to perform such an experiment in miR-146a deficient NF-kB reporter mice, so that the number of BrdU^+^ or Ki67^+^ cells could be compared in the NF-kB + vs − fractions. Such an experiment would provide more direct evidence on behalf of the conclusion, if it is possible to do.

7) Since TRAF6 regulates several pathways (including MAPK and PI3K), are the authors confident that other pathways (beyond NF-kB and IL6) may be important to the phenotype of marrow failure?

8) Since cells are leaving the BM and populating the spleen, it will be useful to analyze the morphology of the BM. Histological analysis of the femur in each genotype miRKO, miR/Rag1,… (as in Figure 7) can provide important insight into the state of the HSC niche.

9) In certain contexts CXCR4 is controlled by miR-146. Given the mislocalization phenotype, it is worth looking at the expression levels of the chemotactic factor, CXCR4.

10) The authors do not include no-LPS controls in Figure 1 and Figure 5. They also do not include wt and mir146a deficient controls in Figure 4.

---

## [Author Response]

*1) The authors summarize a model in the Discussion in which miR-146a acts cell-autonomously to activate NF-kB in HSCs by decreasing TRAF6 expression. This leads to IL-6 expression, which then promotes HSC depletion through increased myeloid differentiation. This is highly speculative and blurs the distinction between what Zhao et al. have tested and what they have not. Is the modest increase in TRAF6 really mediating the effects of miR-146a on HSCs? Does TRAF6 act autonomously in HSCs? What cells secrete IL-6? Does IL-6 really act on HSCs, or on other cells? Can IL-6 really promote myeloid differentiation by HSCs, or does it just promote the proliferation of myeloid progenitors that arise from HSCs? The authors have provided a lot of interesting and compelling data and should not be expected to answer all of these questions in this study; however, they should be careful to distinguish between what they have shown and what is speculation in the Discussion*.

We agree. We have shown that these players (TRAF6, NF-κB, and IL-6) are critical. We have not totally disentangled which molecule is doing what to which cell type. Please see the detailed answers to each of the questions below and the revised manuscript.

*Is the modest increase in TRAF6 really mediating the effects of miR-146a on HSCs*?

We have not fully tested this question. We have only shown enforced TRAF6 expression is sufficient to induce a similar albeit more rapid phenotype. To better answer this question, we will need to knockdown or knockout TRAF6 in miR-146a^-/-^ HSCs to test whether this can rescue the phenotype. Crossing miR-146a KO with TRAF6 KO will take 6 months to generate double KO. If using siRNA to knockdown TRAF6 in miR-146a KO HSCs, this experiment will require at least 6-month follow-up in order to sufficiently assess the status of HSCs. This set of studies would delay the publication of the current manuscript by about a year. We plan to include the more definitive analysis of the miR-146a target in HSCs in the next study.

*Does TRAF6 act autonomously in HSCs*?

We have not shown that TRAF6 acts autonomously in HSCs. We overexpressed TRAF6 in enriched HSPCs, which included HSCs and progenitor cells. In addition, these HSPCs will give rise to downstream mature cells, which also have TRAF6 overexpression. Given the rapid development (in one month) of bone marrow failure phenotype, it is likely that TRAF6 acts directly within the HSPC population. However, whether it’s within HSCs or more differentiated progenitor cells requires further studies.

*What cells secrete IL-6*?

We have shown that miR-146a^-/-^ macrophages and T cells are important producers of IL-6 (Figure 6 and new Figure 7). We have conducted an additional experiment in which we transferred WT, miR-146a KO, and IL-6/miR-146a double KO T cells into Rag1/miR-146a double KO mice to determine whether transfer of miR-146a KO T cells can re-induce HSC depletion and bone marrow failure phenotype (Figure 7). Indeed, we have shown that transfer of miR-146a KO T cells, but not WT or IL-6/miR-146a double KO T cells induced HSC depletion, supporting that IL-6 produced by miR-146a KO T cells is important. This new result has now been added to the revised manuscript.

*Does IL-6 really act on HSCs, or on other cells? Can IL-6 really promote myeloid differentiation by HSCs, or does it just promote the proliferation of myeloid progenitors that arise from HSCs*?

This is a very thoughtful comment. We have conducted additional experiments to test whether IL-6 acts directly on HSCs or more differentiated progenitor cells. In the new Figure 7, we have shown that IL-6 can act directly on miR-146a-/- cKit^+^ cells or sorted LSK cells, but not long-term HSCs, to stimulate increased proliferation and differentiation. This suggests that the direct effect of IL-6 may not be on long-term HSCs, but instead is on the more differentiated short-term HSCs and/or multipotent progenitor cells contained within the LSK populations. We have added the new results and have carefully revised the Discussion accordingly.

*2) Is it really clear that mir-146a is acting cell-autonomously in HSCs? When small numbers of HSCs were transplanted with wild-type marrow, they did not observe HSC depletion in the recipient mice (Figure 2—figure supplement 1), and in competitive whole bone marrow transplants HSC depletion was not observed until 10 months after the transplant (Figure 2). Furthermore, they show that Rag deficiency partially rescues HSC depletion (Figure 4), raising the possibility that engrafted inflammatory T-cells or neoplastic myeloid cells indirectly cause HSC depletion. If only the mir-146a deficient HSCs are depleted in the competitive reconstitution experiments (not the competing wild-type HSCs) then this possibility would seem far-fetched. However, the data are usually presented in unconventional ways for the reconstitution experiments, making it hard to assess whether the competing wild-type HSCs consistently fail to be depleted. It would be very helpful to know absolute numbers of mir-146a deficient versus wild-type HSCs in the same mice*.

This is again a very thoughtful comment. We agree that inflammatory T-cells and/or neoplastic myeloid cells can indirectly cause HSC depletion. We have conducted an additional experiment in which we transferred miR-146a KO T cells into Rag1/miR-146a double KO mice to determine whether transfer of miR-146a KO T cells can re-induce HSC depletion and bone marrow failure phenotype (new Figure 7). Indeed, we have shown that transfer of miR-146a KO T cells induced HSC depletion while WT T cells did not.

In addition, miR-146a is also acting cell-autonomously in HSCs. The evidence is from a competitive bone marrow transplant experiment. As the reviewers pointed out, after 10 months, the ratios of CD45.2/CD45.1 HSCs and LSK cells are less than 0.5 (Figure 2). This means in the same environment, KO HSCs and LSK cells are out-numbered by WT HSCs and LSK cells. Because in WT:KO chimera mice, WT and KO HSCs are experiencing the same insult from WT or KO mature cells or environmental stimuli, if KO HSCs don’t have an intrinsic defect, the ratios between KO and WT HSCs should be 1. The ratios should change only when KO and WT HSCs have intrinsic difference in response to identical external factors. As suggested by the reviewers, we have also included three new figures (new Figure 2) showing the absolute numbers of total, CD45.2^+^ and CD45.1^+^ LSK cells and HSCs in WT:WT and KO:WT chimera mice to make this point more clearly. The new figures show that in the KO:WT chimera mice, KO (CD45.2^+^) LSK cells and HSCs are preferentially depleted while WT (CD45.1^+^) cells remain largely unaffected. If anything, WT HSCs actually show a trend towards increased number, probably as a compensatory mechanism to make up for the loss of KO HSCs in order to restore the normal level of HSCs in the animals.

*3) The authors could more directly test whether NF-kB mediates HSC depletion cell-autonomously by transplanting wild-type and p50-deficient sorted HSCs with mir-146a-deficient competitor marrow. If the donor HSCs become depleted, the effects are likely non-cell autonomous to HSCs and they are an indirect consequence of other mir-146a deficient hematopoietic cells. If p50-deficient HSCs are not depleted even as the mir-146a competitor marrow causes inflammation and MPD, then NF-kb likely causes depletion cell-autonomously as the authors have concluded*.

This is a good suggestion. To set up this experiment, we will have to breed either p50-deficient mice or miR-146a-deficient mice onto a different background (e.g., CD45.1 or actin-GFP mice) in order to distinguish it from the co-transferred competitor marrow. In addition, the analysis is likely to take another 6–12 months. We will include this in a future study. For the current manuscript, we have revised the Discussion to make it clear that we haven’t proven that TRAF6 and NF-κB act cell-autonomously within HSCs to promote HSC depletion.

*4) It is possible that the bone marrow HSCs in mir-146a deficient mice are not being depleted at all but are just being mobilized to the spleen. The authors need to look at the total number of HSCs in the bone marrow and the spleen to determine whether there is actually a decline in the total number of HSCs*.

This is an interesting point. We have analyzed both bone marrow and spleen HSCs from the same 6–7 month-old WT and miR-146a KO mice. Even though there is a several fold increase in spleen HSCs in miR-146a KO mice, the total number of bone marrow and spleen HSCs (LSK CD150^+^CD48^-^) still shows consistent reduction. The depletion is more severe when we apply an additional stem cell marker to define long-term HSCs (LSK CD150^+^CD48^-^EPCR^+^) (Figure 1). Furthermore, we have previously shown that 12–18 month-old miR-146a KO mice develop severe pancytopenia, indicating that expanded splenic hematopoiesis is insufficient to functionally compensate for the defective marrow hematopoiesis (Zhao et al PNAS 2011 and Boldin et al JEM 2011).

*5) Even if there is a decline in the total number of HSCs, it is not clear this reflects “accelerated differentiation” into myeloid cells. The authors cite the increase in LSK cells as evidence for accelerated differentiation. However, there are many alternative possible explanations that have not been addressed. For example, LSK cells might proliferate at a higher rate (Figure 4) or the differentiation or phenotype of cells other than HSCs may be altered. Given that the idea of accelerated HSC differentiation is speculation, the authors should not present it as a tested conclusion*.

This is a good point. We have re-written the relevant part to take out “accelerated differentiation” and replaced it with a more general term “dysregulated hematopoiesis”.

*6) In Figure 5, the authors show (using the NF-kB reporter strain) that ∼2x more miR-146a KO cells have active NF-kB signaling and argue that NF-kB regulates HSC proliferation. Although miR-146a/NF-kB double ko mice have nearly normal numbers of HSCs, it is still indirect evidence that NF-kB regulates HSC proliferation. In Figure 5 the authors show that miR-146a ko cells proliferate more strongly. It would be of interest for the authors to perform such an experiment in miR-146a deficient NF-kB reporter mice, so that the number of BrdU^+^ or Ki67^+^ cells could be compared in the NF-kB + vs − fractions. Such an experiment would provide more direct evidence on behalf of the conclusion, if it is possible to do*.

Again this is a very thoughtful comment. We have done the experiment (Figure 5) and seen some interesting results. There is no correlation between NF-kB activation and proliferation within any stem and progenitor populations. In fact, we don’t think that NF-kB activation within HSCs directly activates proliferation. Whether NF-kB may regulate other aspects of HSC biology, such as differentiation, apoptosis, and trafficking, requires further studies. We have revised the relevant paragraphs to reflect the change.

*7) Since TRAF6 regulates several pathways (including MAPK and PI3K), are the authors confident that other pathways (beyond NF-kB and IL6) may be important to the phenotype of marrow failure*?

This is a good point. We have only shown that TRAF6 may be the relevant target of miR-146a in this context because TRAF6 is de-repressed in miR-146a KO HSCs and overexpression of TRAF6 is sufficient to promote bone marrow failure. It is known that TRAF6 overexpression can activate NF-κB. In addition, we have shown that overexpressing TRAF6 with the same pMIG vector in primary mouse CD4 and CD8 T cells activates NF-κB by EMSA analysis ([41] JEM). But we haven’t analyzed MAPK and PI3K pathway in TRAF6 overexpressing mice and thus cannot exclude the possibility that other pathways beyond NF-κB and IL-6 may also be activated. However, it will take additional long term experiments, such as overexpress TRAF6 in p50^-/-^ cells in bone marrow transplant, to test this conclusively.

*8) Since cells are leaving the BM and populating the spleen, it will be useful to analyze the morphology of the BM. Histological analysis of the femur in each genotype miRKO, miR/Rag1,… (as in Figure 7) can provide important insight into the state of the HSC niche*.

We have included histological analysis of femur bones from all strains of mice, including cohorts of WT, miR KO, p50 KO, p50/miR DKO (Figure 5—figure supplement 2), WT, miR KO, Rag1KO, Rag1/miR DKO (Figure 4) and WT, miR KO, IL6 KO, IL6/miR DKO (Figure 6—figure supplement 1). In general, all DKO bones show significant rescue of marrow fibrosis.

*9) In certain contexts CXCR4 is controlled by miR-146. Given the mislocalization phenotype, it is worth looking at the expression levels of the chemotactic factor, CXCR4*.

We have checked the surface expression of CXCR4 by FACS of WT and miR-146a KO HSCs and have not seen any difference.

*10) The authors do not include no-LPS controls in Figure 1 and Figure 5. They also do not include wt and mir146a deficient controls in Figure 4*.

No-LPS controls are essentially the same as 8–10-week-old WT and miR-146a KO mice in the absence of stimulation, which show no differences between WT and miR-146a KO. We have included the controls for Figure 1 (see Figure 1). However, we did not have no-LPS controls for Figure 5 in the same experiments. Figure 4—figure supplement 1B–D and Figure 5—figure supplement 1 contain basal level of GFP expression of various populations of WT and miR-146a KO mice, and can serve as a baseline level of NF-kB activity in the absence of LPS stimulation.

Figure 4 is simply a re-graph of only the Rag1KO and miR/Rag1 DKO groups from Figure 4 in order to make the comparison, as stated in the figure legend: “For comparison, various cell lineages in spleen of Rag1 KO and miR/Rag1 DKO mice were re-graphed (M).” Because in Figure 4, Rag1KO and miR/Rag1 DKO spleens have significantly fewer cells compared to WT and miRKO spleens, it is difficult to appreciate any differences between Rag1KO and miR/Rag1 DKO. We have revised the figure legend to make it clearer.